# A Unified Framework for Bayesian Optimization under Contextual Uncertainty

**Sebastian Shenghong Tay[1][2], Chuan Sheng Foo[2][3], Daisuke Urano[4],**
**Richalynn Chiu Xian Leong[4], Bryan Kian Hsiang Low[1]**
[1]Department of Computer Science, National University of Singapore
[2]Institute for Infocomm Research (I2R), A*STAR, Singapore
[3]Centre for Frontier AI Research (CFAR), A*STAR, Singapore
[4]Temasek Life Sciences Laboratory, Singapore
`sebastian.tay@u.nus.edu`, `foo_chuan_sheng@i2r.a-star.edu.sg`,
`daisuke@tll.org.sg`, `richalynn@tll.org.sg`, `lowkh@comp.nus.edu.sg`

## Abstract

Bayesian optimization under contextual uncertainty (BOCU) is a family of BO problems in which the learner makes a decision prior to observing the context and must manage the risks involved. Distributionally robust BO (DRBO) is a subset of BOCU that affords robustness against context distribution shift, and includes the optimization of expected values and worst-case values as special cases. By considering the first derivatives of the DRBO objective, we generalize DRBO to one that includes several other uncertainty objectives studied in the BOCU literature such as worst-case sensitivity (and thus notions of risk such as variance, range, and conditional value-at-risk) and mean-risk tradeoffs. We develop a general Thompson sampling algorithm that is able to optimize any objective within the BOCU framework, analyze its theoretical properties, and compare it to suitable baselines across different experimental settings and uncertainty objectives.

## 1 Introduction

Bayesian optimization (BO) is a well-established framework for black-box optimization under the constraint of limited function evaluations (Garnett, 2023) that has seen numerous successes in real-world applications, including hyperparameter tuning (Chen et al., 2018), protein design (Romero et al., 2013), and experimental physics (Duris et al., 2020). While standard BO assumes that the learner has full control over all input variables to the objective function, in many practical scenarios, the learner only has control over a subset of variables (decision variables), while the other variables (context variables) may be randomly determined by the environment. For example, in agriculture, a farmer optimizing plant growth may have control over the amount of water the crops receive, but not the amount of sunlight. A line of research we term *BO under contextual uncertainty* (BOCU) studies such scenarios in which the learner makes a decision prior to observing the context and must therefore manage the uncertainty in the final input due to the uncertainty in the context. This is done by maximizing some *uncertainty objective* that takes the context distribution into account, such as the expected value or the worst-case value with respect to the context. *Distributionally robust BO* (DRBO) is a subset of BOCU that generalizes these uncertainty objectives. However, there are many other uncertainty objectives under the BOCU umbrella that have thus far been seen as adjacent to DRBO, including (conditional) value-at-risk, mean-variance tradeoff, and many others.

This paper unifies several of these disparate works into a single general framework. We show that, by considering the first derivatives of the DRBO objective with respect to the margin, the DRBO framework may be extended to include the above-mentioned adjacent uncertainty objectives and many others. We also subsume in our framework *robust satisficing* (Long et al., 2023), a recently proposed uncertainty objective in the operations research literature that has recently been applied in the context of BO (Saday et al., 2023). By unifying these seemingly different settings, we believe our work allows BOCU problems to be studied in a new and interesting light. Moreover, our framework encompasses a family of novel uncertainty objectives not yet studied in the literature with

their own individual interpretations. Beyond theoretical interest, our unified framework has practical ramifications: we develop a single BO algorithm capable of optimizing any of the uncertainty objectives within the framework which simplifies the process of any practitioner who wishes to optimize different objectives. We provide a regret bound and the conditions under which it achieves sublinear regret, and empirically demonstrate its performance in various experimental settings and uncertainty objectives.

**Related work**. Our work unifies several settings in a line of research studying BO under contextual uncertainty (BOCU). The closest works to ours are those on distributionally robust BO (DRBO) (Kirschner et al., 2020; Nguyen et al., 2020; Husain et al., 2023; Inatsu et al., 2022; Tay et al., 2022), an adaptation of distributionally robust optimization (DRO) in operations research (Rahimian & Mehrotra, 2022) to the BO setting. The DRBO framework captures stochastic BO (Toscano-Palmerin & Frazier, 2022) and robust BO (Bogunovic et al., 2018) which optimize expected values and worst-case values respectively. Other works under the BOCU umbrella optimize for various notions of risk, including (conditional) value-at-risk (Cakmak et al., 2020; Nguyen et al., 2021a;b), quantiles/expectiles (Picheny et al., 2022), and a scalar combination of mean and variance (Iwazaki et al., 2021). Makarova et al. (2021) also consider the mean-variance tradeoff, but do not consider the notion of observable contexts and treat the noisy observations as heteroscedastic. Beland & Nair (2017) and Cakmak et al. (2020) propose very general frameworks whose objectives are arbitrary mappings from distributions to real numbers (termed 'robustness metrics' and 'risk measures' respectively); however, they assume the context is within the learner's control during the learning procedure, and the large generality of their frameworks appears to preclude meaningful regret analyses. Our framework extends DRBO in a manner that increases its generality while still being amenable to analysis.

BOCU can be considered to generalize a line of work referred to as contextual BO in which the context is observed *prior* to making a decision (Krause & Ong, 2011; Char et al., 2019; Fiducioso et al., 2019; Kirschner & Krause, 2019), by viewing it as a special case of DRBO in which the reference distribution is a point mass corresponding to the context at that iteration and the margin $\epsilon = 0$. BOCU is a subset of an even more general class of BO works that study optimization under an inability to fully control every decision due to random or adversarial influences, including BO with uncertain inputs (Oliveira et al., 2019), BO for finding equilibria in games (Marchesi et al., 2020; Tay et al., 2023), BO with partially-specified decisions (Hayashi et al., 2022), and causal BO (Aglietti et al., 2020; Sussex et al., 2023), among many others.

## 2 PRELIMINARIES

We study problems involving *decision variables* $\mathbf{x} \in \mathcal{X} \subset \mathbb{R}^m$, *context variables* $\mathbf{c} \in \mathcal{C} = \{\mathbf{c}^{(1)}, \mathbf{c}^{(2)}, ..., \mathbf{c}^{(n)}\} \subset \mathbb{R}^\ell, |\mathcal{C}| = n$, and functions over the joint decision-context space $f : \mathcal{X} \times \mathcal{C} \to \mathbb{R}$. We assume that $\mathcal{X}$ is a compact and convex subset of $\mathbb{R}^m$, and that $\mathcal{C}$ is a finite set (a standard assumption in the DRBO literature (Inatsu et al., 2022; Kirschner et al., 2020; Tay et al., 2022)). The decision variables $\mathbf{x}$ can be chosen by the learner, while the context variables $\mathbf{c}$ are sampled randomly from some probability distribution. The learner chooses $\mathbf{x}$ prior to observing the context $\mathbf{c}$. We assume that some *reference distribution* $\mathbf{p}$ is known that captures the learner's prior knowledge of the distribution governing $\mathbf{c}$. Since $\mathcal{C}$ is finite, $\mathbf{p}$ (and all distributions over $\mathcal{C}$) can be represented as a probability vector in $\mathbb{R}^n$, i.e., $p_i$, the $i$-th entry of $\mathbf{p}$, is the probability that the random vector $\mathbf{c}$ has value $\mathbf{c}^{(i)}$. The learner also chooses some convex distribution distance $d$, e.g., one of the $f$-divergences or integral probability metrics. This work is concerned with optimization problems of the form $\mathbf{x}^* = \arg\max_{\mathbf{x} \in \mathcal{X}} g(\mathbf{x}; f, \mathbf{p}, d, \epsilon)$ where $g$ is a scalar function and $\epsilon$ denotes additional problem-dependent parameters. We refer to $g$ as an *uncertainty objective*. We sometimes omit the parameters $f, \mathbf{p}$ and $d$ to reduce clutter and introduce specific parameters in $\epsilon$ as the need arises. The first examples of such uncertainty objectives are the *stochastic optimization* (SO) (Toscano-Palmerin & Frazier, 2022) and *robust optimization* (RO) (Bogunovic et al., 2018) objectives:

$$g_{\text{SO}}(\mathbf{x}) \coloneqq \mathbb{E}_{\mathbf{c} \sim \mathbf{p}} [f(\mathbf{x}, \mathbf{c})] = \mathbf{p}^\top \mathbf{f}(\mathbf{x}), \qquad g_{\text{RO}}(\mathbf{x}) \coloneqq \min_{\mathbf{c} \in \mathcal{C}} f(\mathbf{x}, \mathbf{c}) \tag{1}$$

where $\mathbf{f}(\mathbf{x}) \coloneqq (f(\mathbf{x}, \mathbf{c}^{(i)}))_{i=1}^n \in \mathbb{R}^n$. Throughout this work, we use bold symbols to represent vectors, and bold symbols followed by $(\mathbf{x})$ to indicate a vector in $\mathbb{R}^n$ constructed by applying a function to each of $(\mathbf{x}, \mathbf{c}^{(1)}), (\mathbf{x}, \mathbf{c}^{(2)}), ..., (\mathbf{x}, \mathbf{c}^{(n)})$. While the SO objective is a straightforward

way to handle contextual uncertainty, it has two shortcomings: first, its correctness is dependent on $\mathbf{p}$ being the true distribution from which $\mathbf{c}$ is sampled; second, the expected value says nothing about how bad individual realizations of $f(\mathbf{x}, \mathbf{c})$ may be, a fact which risk-averse learners may be dissatisfied with. RO is more risk-averse but may be too pessimistic as it does not incorporate any knowledge about the distribution of $\mathbf{c}$ and simply assumes that the worst possible value is attained. *Distributionally robust optimization* (DRO) is a generalization that allows an 'interpolation' between these two objectives (Kirschner et al., 2020; Nguyen et al., 2020; Tay et al., 2022):

$$g_{\text{DRO}}(\mathbf{x}; \epsilon) = v(\mathbf{x}, \epsilon) \coloneqq \min_{\mathbf{q} \in \mathcal{U}(\mathbf{p}, d, \epsilon)} \quad \mathbf{q}^\top \mathbf{f}(\mathbf{x}) \tag{2}$$

$$\mathcal{U}(\mathbf{p}, d, \epsilon) \coloneqq \{\mathbf{q} \in \mathbb{R}^n : \mathbf{q} \succeq \mathbf{0}, \mathbf{1}^\top \mathbf{q} = 1, d(\mathbf{p}, \mathbf{q}) \le \epsilon\}$$

where the *margin* $\epsilon \ge 0$, and $\mathcal{U}(\mathbf{p}, d, \epsilon)$ is the *uncertainty set* and contains elements that are valid probability distributions $\epsilon$-close to the reference distribution $\mathbf{p}$ in terms of $d$. Intuitively, DRO considers how poor the expected value may be if an adversary were allowed to shift the context distribution a distance $\epsilon$ away from $\mathbf{p}$. In this work, we consider only distances $d$ such that $d(\mathbf{p}, \cdot)$ is a convex function of $\mathbf{q}$. The uncertainty set constraints are then convex, and $v(\mathbf{x}, \epsilon)$ is the optimal value of a convex optimization problem. $v(\mathbf{x}, \epsilon)$, also termed the *worst-case expected value* (under $\epsilon$-distribution shift), is non-increasing in $\epsilon$, and is itself a convex function of $\epsilon$ (Boyd & Vandenberghe, 2004, Sec. 5.6). SO is recovered when $\epsilon = 0$, and RO is recovered when $\epsilon \ge \hat{\epsilon}_{\mathbf{x}} \coloneqq d(\mathbf{p}, \mathbf{e}_{\mathbf{x}})$ where $\mathbf{e}_{\mathbf{x}}$ is a probability vector with zeroes everywhere except at an index in $\arg\min_i f(\mathbf{x})_i$. The learner thus chooses their level of risk aversion via $\epsilon$. Fig. 1 illustrates $v(\mathbf{x}, \epsilon)$ as a function of $\epsilon$ and the relationship between SO, RO and DRO.

# 3 A GENERAL FRAMEWORK WITH FIRST DERIVATIVES

We have seen thus far that $v(\mathbf{x}, \epsilon)$ encapsulates the uncertainty objectives SO, RO, and DRO depending on the value of $\epsilon$. It turns out that, by including information about the first derivatives of $v(\mathbf{x}, \epsilon)$ with respect to $\epsilon$, we can include even more uncertainty objectives that have been studied in the BOCU literature. Define the right derivative of $v(\mathbf{x}, \epsilon)$ at $\epsilon$ as

$$\delta(\mathbf{x}, \epsilon) \coloneqq \lim_{h \downarrow 0} \frac{v(\mathbf{x}, \epsilon + h) - v(\mathbf{x}, \epsilon)}{h} . \tag{3}$$

Since $v(\mathbf{x}, \epsilon)$ is a convex function of $\epsilon$ with domain $[0, \infty)$, $\delta(\mathbf{x}, \epsilon)$ exists and is finite everywhere on the open interval $(0, \infty)$ (Rockafellar, 1997, Thm. 23.4), and is non-positive since $v(\mathbf{x}, \epsilon)$ is non-increasing in $\epsilon$. Special care must be taken to ensure that $\delta(\mathbf{x}, 0)$ is finite[1] with a particular choice of distribution distance $d$; distribution distances that fulfill this condition include the total variation (TV) distance, maximum mean discrepancy (MMD), and the Wasserstein metric, among others (Gotoh et al., 2020; Staib & Jegelka, 2019). When $\delta(\mathbf{x}, 0)$ is finite, it is the *worst-case sensitivity* (WCS) uncertainty objective (Gotoh et al., 2020; Tay et al., 2022):

$$g_{\text{WCS}}(\mathbf{x}) \coloneqq \delta(\mathbf{x}, 0) . \tag{4}$$

The worst-case sensitivity may be considered an uncertainty objective as Gotoh et al. (2020) showed that the worst-case sensitivity corresponds to different 'notions of risk' depending on $d$. For example, when $d$ is the TV distance, $\delta(\mathbf{x}, 0)$ is the range $\max_i f(\mathbf{x})_i - \min_j f(\mathbf{x})_j$ (assuming $\mathbf{p}$ has nonzero mass on the indices attaining the maximum and minimum); when $d$ is the maximum mean discrepancy (MMD), $\delta(\mathbf{x}, 0)$ is a quantity closely related to the variance $\mathbb{V}_{\mathbf{c} \sim \mathbf{p}}[f(\mathbf{x}, \mathbf{c})]$ (Staib & Jegelka, 2019); when $d$ is a specific $f$-divergence, $\delta(\mathbf{x}, 0)$ is linked to the conditional value-at-risk (Nguyen et al., 2021a). Just as DRO may be interpreted as an interpolation between SO and RO, in order to interpolate between the expected value (SO) and these notions of risk (WCS), a learner may choose as their uncertainty objective a scalar combination of the expected value and these notions of risk (Iwazaki et al., 2021), which we refer to as the *mean-risk tradeoff* (MR) with tradeoff parameter $\beta > 0$:

$$g_{\text{MR}}(\mathbf{x}; \beta) \coloneqq \mathbb{E}_{\mathbf{c} \sim \mathbf{p}}[f(\mathbf{x}, \mathbf{c})] + \beta\delta(\mathbf{x}, 0) . \tag{5}$$

MR as a function of $\beta$ is the linearization of $v(\mathbf{x}, \epsilon)$ around $\epsilon = 0$ (see Fig. 1). By the convexity of $v(\mathbf{x}, \epsilon)$ in $\epsilon$, $g_{\text{MR}}(\mathbf{x}; \beta) \le v(\mathbf{x}, \beta)$ for all $\beta \ge 0$.

---

[1] For example, $\delta(\mathbf{x}, 0) = \infty$ if $d$ belongs to a particular family of smooth $f$-divergences (Gotoh et al., 2020).

Table 1: Various uncertainty objectives studied in the literature, their expressions, and the parameters required to recover them from our general uncertainty objective Equation 7. A bolded objective indicates that all objectives indented under it are special cases.

| Uncertainty objective | Expression | $\alpha$ | $\beta$ | $\epsilon$ |
|---|---|---|---|---|
| **Dist. robust opt. (DRO)** (2) | $v(\mathbf{x}, \epsilon) := \min\limits_{\mathbf{q} \in \mathcal{U}(\mathbf{p}, d, \epsilon)} \mathbb{E}_{\mathbf{c} \sim \mathbf{q}} [f(\mathbf{x}, \mathbf{c})]$ | 1 | 0 | $[0, \infty)$ |
| Stochastic opt. (SO) (1) | $\mathbb{E}_{\mathbf{c} \sim \mathbf{p}} [f(\mathbf{x}, \mathbf{c})]$ | 1 | 0 | 0 |
| Robust opt. (RO) (1) | $\min\limits_{\mathbf{c} \in \mathcal{C}} f(\mathbf{x}, \mathbf{c})$ | 1 | 0 | $[\hat{\epsilon}_{\mathbf{x}}, \infty)$ |
| **Robust satisficing (RS)** (6) | $\max\limits_{k : k \leq 0} k \text{ s.t. } \forall \epsilon, v(\mathbf{x}, \epsilon) \geq \tau_{\mathbf{x}} + k\epsilon$ | 0 | 1 | $[0, \infty)$ |
| Worst-case sensitivity (WCS) (4) | $\delta(\mathbf{x}, 0) := \lim\limits_{h \downarrow 0} \dfrac{v(\mathbf{x}, \epsilon + h) - v(\mathbf{x}, \epsilon)}{h}$ | 0 | 1 | 0 |
| Mean-risk tradeoff (MR) (5) | $\mathbb{E}_{\mathbf{c} \sim \mathbf{p}} [f(\mathbf{x}, \mathbf{c})] + \beta\delta(\mathbf{x}, 0)$ | 1 | $[0, \infty)$ | 0 |

The uncertainty objectives WCS and MR only use the quantity $\delta(\mathbf{x}, 0)$. A natural question arises: are there uncertainty objectives that rely on $\delta(\mathbf{x}, \epsilon)$ for some $\epsilon > 0$? *Robust satisficing* (Long et al., 2023; Saday et al., 2023), a recently introduced concept in the operations research literature, is one such objective:

$$g_{\mathrm{RS}}(\mathbf{x}; \tau) := \max\limits_{k : k \leq 0} k \text{ s.t. } \forall \epsilon, v(\mathbf{x}, \epsilon) \geq \tau + k\epsilon \tag{6}$$

where $\tau \leq \mathbb{E}_{\mathbf{c} \sim \mathbf{p}} [f(\mathbf{x}, \mathbf{c})]$, otherwise the constraint is violated at $\epsilon = 0$ for all $k$. In robust satisficing, the learner chooses the parameter $\tau$ that represents a 'good enough' expected value under the reference distribution. The quantity $g_{\mathrm{RS}}(\mathbf{x}; \tau)$, also known as the *antifragility*, is the maximum non-positive slope such that the linear function $\tau + g_{\mathrm{RS}}(\mathbf{x}; \tau)\epsilon$ lower bounds $v(\mathbf{x}, \epsilon)$ for all $\epsilon \geq 0$. The interpretation is as follows: the learner accepts some baseline expected value $\tau$, and in the case of distribution shift, accepts a decrease in the expected value proportional to the distance between the reference and true distributions; the decision $\mathbf{x}$ that minimizes this decrease per unit distance is considered most robust. Fig. 1 illustrates the relationship between $\tau$, $v(\mathbf{x}, \epsilon)$, and the robust satisficing objective (the gradient of the dotted line). The following proposition (proofs of all results are in Appendix A) establishes the link between robust satisficing and the first derivatives of $v(\mathbf{x}, \epsilon)$.

**Proposition 3.1.** *Suppose the distribution distance $d$ is such that $\delta(\mathbf{x}, 0)$ is finite. Fix a $\tau \leq \min_{\mathbf{x} \in \mathcal{X}} \mathbb{E}_{\mathbf{c} \sim \mathbf{p}} [f(\mathbf{x}, \mathbf{c})]$. For all $\mathbf{x} \in \mathcal{X}$, there exists an $\epsilon_{\mathbf{x}} \geq 0$ such that $g_{\mathrm{RS}}(\mathbf{x}; \tau) \in \partial v(\mathbf{x}, \epsilon_{\mathbf{x}})$ where $\partial v(\mathbf{x}, \epsilon_{\mathbf{x}})$ is the subdifferential of $v(\mathbf{x}, \epsilon)$ at $\epsilon_{\mathbf{x}}$.*

*Conversely, fix an $\epsilon \geq 0$. For all $\mathbf{x} \in \mathcal{X}$, there exists a $\tau_{\mathbf{x}} \leq \mathbb{E}_{\mathbf{c} \sim \mathbf{p}} [f(\mathbf{x}, \mathbf{c})]$ such that $\delta(\mathbf{x}, \epsilon) = g_{RS}(\mathbf{x}; \tau_{\mathbf{x}})$. In particular, $g_{WCS}(\mathbf{x}) = \delta(\mathbf{x}, 0) = g_{RS} (\mathbf{x}; \mathbb{E}_{\mathbf{c} \sim \mathbf{p}} [f(\mathbf{x}, \mathbf{c})])$.*

In other words, the robust satisficing objective and the subderivatives of $v(\mathbf{x}, \epsilon)$ are closely related. For a fixed $\tau$, the robust satisficing objective is a subderivative of $v(\mathbf{x}, \epsilon)$ at some value of $\epsilon = \epsilon_{\mathbf{x}}$ (which may be different for each decision $\mathbf{x}$); likewise, for a fixed $\epsilon$, the right derivative at $\epsilon$ (which is a subderivative) is the robust satisficing objective for some $\tau = \tau_{\mathbf{x}}$ (which again may be different for each decision $\mathbf{x}$). This close link suggests that the right derivative $\delta(\mathbf{x}, \epsilon)$ at values of $\epsilon$ other than 0 can also be considered uncertainty objectives. This aligns with intuition: the smaller the magnitude of the (always non-positive) gradient at a certain value of $\epsilon$, the smaller the potential decrease in $v(\mathbf{x}, \epsilon)$ at larger values of $\epsilon$ by the convexity of $v(\mathbf{x}, \epsilon)$.

### 3.1 GENERAL UNCERTAINTY OBJECTIVE

The discussion thus far suggests that both $v(\mathbf{x}, \epsilon)$ and $\delta(\mathbf{x}, \epsilon)$ are useful quantities in the study of BO under contextual uncertainty. We incorporate both quantities and introduce the scaling parameters $\alpha \geq 0$ and $\beta \geq 0$ to obtain the general uncertainty objective

$$g(\mathbf{x}; \alpha, \beta, \epsilon) = \alpha v(\mathbf{x}, \epsilon) + \beta\delta(\mathbf{x}, \epsilon). \tag{7}$$

Table 1 shows the values of parameters $\alpha$, $\beta$ and $\epsilon$ required to recover the uncertainty objectives described in this work. With a suitable choice of parameters, $g(\mathbf{x}; \alpha, \beta, \epsilon)$ recovers all of them

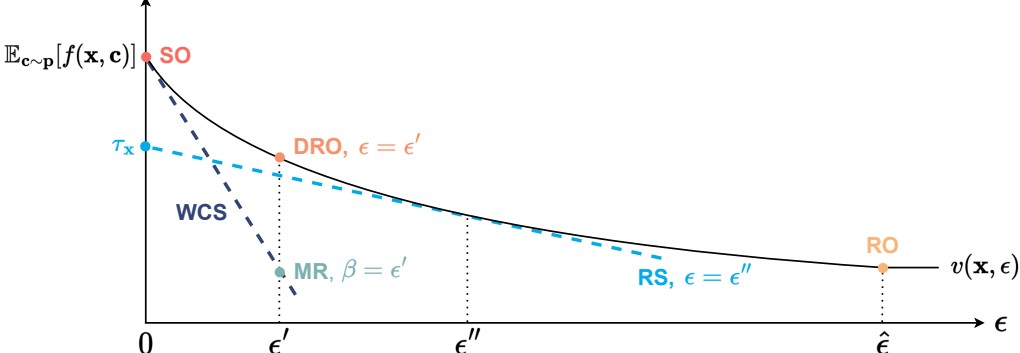

Figure 1: Diagram illustrating the convex function $v(\mathbf{x}, \epsilon)$ for some decision $\mathbf{x}$, and how the various uncertainty objectives relate to the function. The values of DRO, SO, RO, and MR are their dots' vertical coordinates, while the values of RS and WCS are the gradients of their dotted lines.

exactly with the exception of robust satisficing, which has the following caveat: from Prop. 3.1, with a fixed choice of $\epsilon$, $g(\mathbf{x}; 0, 1, \epsilon) = \delta(\mathbf{x}, \epsilon) = g_{\text{RS}}(\mathbf{x}; \tau_{\mathbf{x}})$, which is the robust satisficing objective for a *possibly different value of $\tau_{\mathbf{x}}$ for each decision* $\mathbf{x}$. A practitioner who wishes to optimize the robust satisficing objective with the same $\tau$ for every decision $\mathbf{x}$ using our general objective must know *a priori* the specific $\epsilon_{\mathbf{x}}$ for each $\mathbf{x}$ that corresponds to $\tau$. This is generally not possible since $f$ is assumed to be unknown. Nevertheless, optimizing $g(\mathbf{x}; 0, 1, \epsilon)$ still corresponds to optimizing the robust satisficing objective, only with a different $\tau_{\mathbf{x}}$ for each $\mathbf{x}$.

Our general objective also allows for the optimization of novel uncertainty objectives that have not been explicitly studied so far. The learner is free to choose values of $\alpha$, $\beta$ and $\epsilon$ that do not recover previously studied uncertainty objectives and instead result in an uncertainty objective with alternative interpretations as desired. For example, choosing $\alpha, \beta, \epsilon$ to all be greater than $0$ results in an uncertainty objective that favors decisions with large worst-case expected values under $\epsilon$-distribution shift ($v(\mathbf{x}, \epsilon)$ term), and that also do not decrease in worst-case expected value too much as $\epsilon$ increases ($\delta(\mathbf{x}, \epsilon)$ term). Such an objective would be similar to MR except that the objective is calculated at values of $\epsilon$ other than $0$, which more explicitly accounts for distribution shift. This general uncertainty objective may also be used if the learner wishes to maximize an 'interpolation' of DRO and RS objectives with the level of interpolation decided by the relative values of $\alpha$ and $\beta$. We observe experimentally (see Appendix B) that BO algorithms that maximize an objective with $\alpha, \beta$ greater than $0$ are also 'robust' to the choice of uncertainty objective in that they also have relatively good performance under both DRO ($\alpha = 1, \beta = 0$) and RS ($\alpha = 0, \beta = 1$) objectives.

## 4 A GENERAL ALGORITHM

We present a general BO acquisition function based on Thompson sampling (Thompson, 1933; Russo & Van Roy, 2014) that optimizes the general uncertainty objective Equation 7 for any valid choices of $\alpha$, $\beta$ and $\epsilon$, and is thus capable of optimizing any of the uncertainty objectives in the BOCU literature covered in this work.

We first formalize the learning setting. We assume the unknown function $f$ is sampled from a Gaussian process (GP) with mean $0$ and kernel (covariance function) $k(\mathbf{x}, \mathbf{c}; \mathbf{x}', \mathbf{c}')$, with $k(\mathbf{x}, \mathbf{c}; \mathbf{x}, \mathbf{c}) = 1$. At each iteration $t$, the learner chooses a decision $\mathbf{x}_t \in \mathcal{X}$ while $\mathbf{c}_t \in \mathcal{C}$ is sampled from $\mathbf{p}_t^*$, and the learner receives a noisy observation $y_t = f(\mathbf{x}_t, \mathbf{c}_t) + \xi_t$, where each $\xi_t$ is i.i.d. noise sampled from the Gaussian distribution $\mathcal{N}(0, \sigma^2)$. We assume that the sequence of true distributions $\{\mathbf{p}_t^*\}_{t=1}^T$ is fixed but unknown to the learner. Under these assumptions, the posterior mean and variance of $f$ at a decision-context pair $(\mathbf{x}, \mathbf{c})$ at the start of iteration $t$ (having had $t-1$ observations) are

$$\mu_{t-1}(\mathbf{x}, \mathbf{c}) = \mathbf{k}_{t-1}(\mathbf{x}, \mathbf{c})^\top (\mathbf{K}_{t-1} + \sigma^2 \mathbf{I})^{-1} \mathbf{y}_{t-1}$$

$$\sigma_{t-1}^2(\mathbf{x}, \mathbf{c}) = k(\mathbf{x}, \mathbf{c}; \mathbf{x}, \mathbf{c}) - \mathbf{k}_{t-1}(\mathbf{x}, \mathbf{c})^\top (\mathbf{K}_{t-1} + \sigma^2 \mathbf{I})^{-1} \mathbf{k}_{t-1}(\mathbf{x}, \mathbf{c}) \quad (8)$$

where $k_{t-1,i}(\mathbf{x}, \mathbf{c}) \coloneqq k(\mathbf{x}, \mathbf{c}; \mathbf{x}_i, \mathbf{c}_i)$, and $K_{t-1,ij} \coloneqq k(\mathbf{x}_i, \mathbf{c}_i; \mathbf{x}_j, \mathbf{c}_j)$. Finally, the learner has their choice of uncertainty objective parameters $(\alpha, \beta, \{\epsilon_t\}_{t=1}^T)$ and reference distributions $\{\mathbf{p}_t\}_{t=1}^T$.

---

**Algorithm 1** TS-BOCU

---

1: **Input:** GP with kernel $k$, maximum iteration $T$, uncertainty objective parameters $(\alpha, \beta, \{\epsilon_t\}_{t=1}^T)$, reference distributions $\{\mathbf{p}_t\}_{t=1}^T$
2: **for** iteration $t = 1$ **to** $T$ **do**
3:      Sample model $\tilde{f}_t$ from GP posterior given $\mathcal{D}_{t-1}$
4:      Select $\mathbf{x}_t := \arg\max_{\mathbf{x} \in \mathcal{X}} \tilde{g}(\mathbf{x}; \tilde{f}_t, \mathbf{p}_t, \alpha, \beta, \epsilon_t)$
5:      Observe $\mathbf{c}_t \sim \mathbf{p}_t^*$
6:      Observe $y_t = f(\mathbf{x}_t, \mathbf{c}_t) + \xi_t, \xi_t \sim \mathcal{N}(0, \sigma^2)$
7:      $\mathcal{D}_t := \{(\mathbf{x}_\tau, \mathbf{c}_\tau, y_\tau)\}_{\tau=1}^t$
8: **end for**
9: **return** $\mathcal{D}_T$

---

The proposed acquisition, named TS-BOCU (Thompson sampling for BO under contextual uncertainty), is described in Algorithm 1. As with all Thompson sampling-based algorithms, TS-BOCU relies on a function $\tilde{f}_t$ sampled from the GP posterior at iteration $t$. The surrogate objective $\tilde{g}(\mathbf{x}; \tilde{f}_t, \mathbf{p}_t, \alpha, \beta, \epsilon_t)$ is the same as the general uncertainty objective Equation 7, except that the unknown true function $f$ is replaced with the known sampled function $\tilde{f}_t$, and we make the dependence on $\tilde{f}_t$ and $\mathbf{p}_t$ explicit. $\tilde{g}$ can then be computed by computing $v(\mathbf{x}, \epsilon)$ and $\delta(\mathbf{x}, \epsilon)$ (which relies on two computations of $v)^2$, and $v$ in turn can be computed by solving the convex optimization problem Equation 2. The learner chooses the decision $\mathbf{x}_t$ that maximizes this surrogate objective.

## 4.1 THEORETICAL ANALYSIS

We analyze the $T$-period Bayesian cumulative regret (Russo & Van Roy, 2014) of TS-BOCU:

$$\text{BayesRegret}(T) = \sum_{t=1}^T \mathbb{E}\left[g_t(\mathbf{x}_t^*) - g_t(\mathbf{x}_t)\right]$$

where $g_t(\mathbf{x}) := g(\mathbf{x}; f, \mathbf{p}_t, \alpha, \beta, \epsilon_t)$ (making the dependence on $f$ and $\mathbf{p}_t$ explicit), $\mathbf{x}_t^* := \arg\max_{\mathbf{x} \in \mathcal{X}} g_t(\mathbf{x})$, and the expectation is taken over the prior distribution of the underlying function $f$ and the random outcomes of the experiment formalized as the *history* $H_T := (\mathbf{x}_1, \mathbf{c}_1, y_1, ..., \mathbf{x}_{T-1}, \mathbf{c}_{T-1}, y_{T-1}, \mathbf{x}_T, \mathbf{c}_T)$. Our main result bounds the Bayesian regret incurred by TS-BOCU and provides the conditions under which the algorithm achieves a sublinear regret.

**Theorem 4.1.** *If the following assumptions hold:*

1. *The kernel $k$ is the squared exponential kernel or a Matérn kernel with $\nu > 2$;*
2. *the distribution distance $d$ is the total variation (TV) distance;*
3. *for all $t \leq T$, the true distribution $\mathbf{p}_t^*$ satisfies the following:*
      *(a) $d(\mathbf{p}_t, \mathbf{p}_t^*) \leq \epsilon_t$;*
      *(b) for all $c \in [|\mathcal{C}|]$, $p_{t,c}^* \geq p_{min} > 0$,*

*then TS-BOCU (Algorithm 1) incurs a $T$-period Bayesian regret bounded by*

$$\text{BayesRegret}(T) \leq \mathcal{O}\Big(\alpha\Big(\sqrt{T(A_1\gamma_T + A_2)\ln(T^2\tau_t^m n)} + \sum_{t=1}^T (\mathbb{E}[B\epsilon_t])\Big)$$
$$+ \beta\Big(\sqrt{\ln(T^2\tau_t^m n)} + \sqrt{\ln n} + p_{min}^{-1}\sqrt{T(A_1\gamma_T + A_2)\ln(T^2\tau_t^m n)}\Big)\Big) \quad (9)$$

*where $\gamma_T := \max_{\{(\mathbf{x}_t, \mathbf{c}_t)\}_{t=1}^T}(1/2)\ln|\mathbf{I} + \sigma^{-2}\mathbf{K}_T|$ is the maximum information gain, $B := \max_{\mathbf{x} \in \mathcal{X}} \|\mathbf{f}(\mathbf{x})\|$, $\tau_t := t^2 mab$, $A_1 := \ln(1 + \sigma^{-2})^{-1}$, $A_2 := \frac{1}{6}Lm^2ab\pi^{5/2}$, and $L$, $a$ and $b$ are kernel-dependent constants.*

The proof of Theorem 4.1 adapts proof techniques from Russo & Van Roy (2014) for bounding the Bayesian regret of Thompson sampling algorithms and from Kirschner et al. (2020) for the extension

---

$^2$In practice, to approximate taking the limit $h \downarrow 0$ in Equation 3 when computing $\delta(\mathbf{x}, \epsilon)$, we choose a $h$ as small as possible while still retaining numerical stability.

to the DRBO setting. Our analysis focuses on bounding the regret incurred due to the new derivative term in the general uncertainty objective Equation 7. The proof relies on an upper confidence bound sequence $U_t$ of the form

$$U_t(\mathbf{x}) := \alpha \hat{v}_t(\mathbf{x}) + \beta \hat{\delta}_t(\mathbf{x}) \tag{10}$$

$$\hat{v}_t(\mathbf{x}) := \min_{\mathbf{q} \in \mathcal{U}(\mathbf{p}_t, d, \epsilon_t)} \mathbf{q}^\top \mathbf{u}_t(\mathbf{x})$$

$$\hat{\delta}_t(\mathbf{x}) := \max_{\boldsymbol{\ell}_t(\mathbf{x}) \preceq \boldsymbol{\varphi} \preceq \mathbf{u}_t(\mathbf{x})} \lim_{h \to 0} \frac{1}{h} \left( \min_{\mathbf{q} \in \mathcal{U}(\mathbf{p}_t, d, \epsilon_t + h)} \mathbf{q}^\top \boldsymbol{\varphi} - \min_{\mathbf{q} \in \mathcal{U}(\mathbf{p}_t, d, \epsilon_t)} \mathbf{q}^\top \boldsymbol{\varphi} \right) \tag{11}$$

$$\mathbf{u}_t(\mathbf{x}) := \boldsymbol{\mu}_{t-1}(\mathbf{x}) + \sqrt{\beta_t} \boldsymbol{\sigma}_{t-1}(\mathbf{x}), \quad \boldsymbol{\ell}_t(\mathbf{x}) := \boldsymbol{\mu}_{t-1}(\mathbf{x}) - \sqrt{\beta_t} \boldsymbol{\sigma}_{t-1}(\mathbf{x})$$

We verify that $U_t$ is well-defined under the assumptions of Theorem. 4.1:

**Proposition 4.2.** *If the distribution distance $d$ is the total variation (TV) distance, then the derivative upper confidence bound $\hat{\delta}_t(\mathbf{x})$ defined in Equation 11 exists.*

**Conditions for sublinear regret.** The maximum information gain $\gamma_T$ in Equation 9 is a kernel-dependent quantity that is standard in BO regret bound analysis (Srinivas et al., 2010). As long as $\gamma_T < \mathcal{O}(T)$ (as is the case for the popular squared exponential kernel), all terms in the regret bound are sublinear in $T$ with the possible exception of the $\sum_{t=1}^T \mathbb{E}[B\epsilon_t]$ term. This term also appears in the analysis of Kirschner et al. (2020) for DRBO under the 'general' setting, and can be guaranteed to be sublinear with additional assumptions. For example (under the 'data-driven' setting of Kirschner et al. (2020)), if $\mathbf{p}_t^*$ is a constant $\mathbf{p}^*$ for all $t$, $\epsilon_t = d(\mathbf{p}_t, \mathbf{p}^*)$, and $\mathbf{p}_t$ is the maximum likelihood estimate of $\mathbf{p}^*$ given $\{\mathbf{c}_i\}_{i=1}^{t-1}$, then $\mathbf{p}_t \to \mathbf{p}^*$ as $t \to \infty$ with probability 1 (Rao, 1957), implying that $\epsilon_t \to 0$ with probability 1 and thus that $\sum_{t=1}^T \mathbb{E}[\epsilon_t]$ is sublinear. $B$ is a function of $f$ that can be bound as a constant that does not depend on $T$, and is a random variable independent of the sequence $\{\epsilon_t\}$ generated in the aforementioned manner. We thus conclude that $\sum_{t=1}^T \mathbb{E}[B\epsilon_t] = \mathbb{E}[B] \sum_{t=1}^T \mathbb{E}[\epsilon_t]$ is sublinear under the additional assumptions above.

**Necessity of assumptions.** Under the assumption that the kernel $k$ is the squared exponential kernel or a Matérn kernel with $\nu > 2$, the derivatives of GP sample paths $f$ satisfy $\mathbb{P}(\sup_{(\mathbf{x},\mathbf{c}) \in \mathcal{X} \times \mathcal{C}} |\frac{\partial f(\mathbf{x},\mathbf{c})}{\partial x_i}| > J) \leq ae^{-(J/b)^2}$ for all $i \in \{1, ..., m\}$ and kernel-dependent constants $a$ and $b$ (Ghosal & Roy, 2006). This high probability bound is a standard requirement (Kandasamy et al., 2018; Srinivas et al., 2010) that enables analysis of continuous decision sets via an iterative discretization argument. The assumption that $d$ is the TV distance aids analysis considerably as TV admits a closed form for the optimal solution of the convex optimization problem Equation 2 (see Lemma A.11 in Appendix A.11). The derivative term presents significant challenges: the limit term causes difficulties without a closed form, and the inner term is a difference of worst-case expected values which, informally speaking, 'erases' the dependency on the uncertainty set and precludes bounding it in terms of the margin $\epsilon$ as was done for the $v(\mathbf{x}, \epsilon)$ term. The proof of Theorem 4.1 that bounds the regret incurred due to the derivative term thus relies on the closed form induced by TV several times. Also, for Proposition 4.2, showing the existence of $\hat{\delta}_t(\mathbf{x})$ for general $d$ involves proving the uniform convergence of the limit term, a task that is again aided by the closed form induced by TV. Despite this assumption, we show empirically in Sec. 5 that TS-BOCU displays the sublinear property even when $d$ is the MMD. We thus conjecture that a similar result exists for some general class of distances and leave the proof of such an extended result to future work.

**Thompson sampling enables tractability.** Interestingly, the general objective can be optimized easily with Thompson sampling, whereas it is unclear whether there are UCB-based acquisitions (Srinivas et al., 2010; Chowdhury & Gopalan, 2017; Kirschner et al., 2020) that are both tractable and theoretically justified. The upper confidence bound sequence $U_t$ in Equation 10 is difficult to compute as $\hat{\delta}_t$ is the solution of a bilevel optimization problem. UCB-based acquisitions require an explicit computation of the upper confidence bound, whereas Thompson sampling does not require it and only requires the ability to sample from the posterior. In fact, the Bayesian regret of Thompson sampling holds simultaneously for *all possible choices* of upper confidence bound sequence $U_t$ and thus performs as well as the *best* choice of $U_t$ would predict (Russo & Van Roy, 2014). The BOCU framework presented in this work is an interesting example of a family of BO problems in which Thompson sampling is significantly easier to run than its UCB-based counterparts.

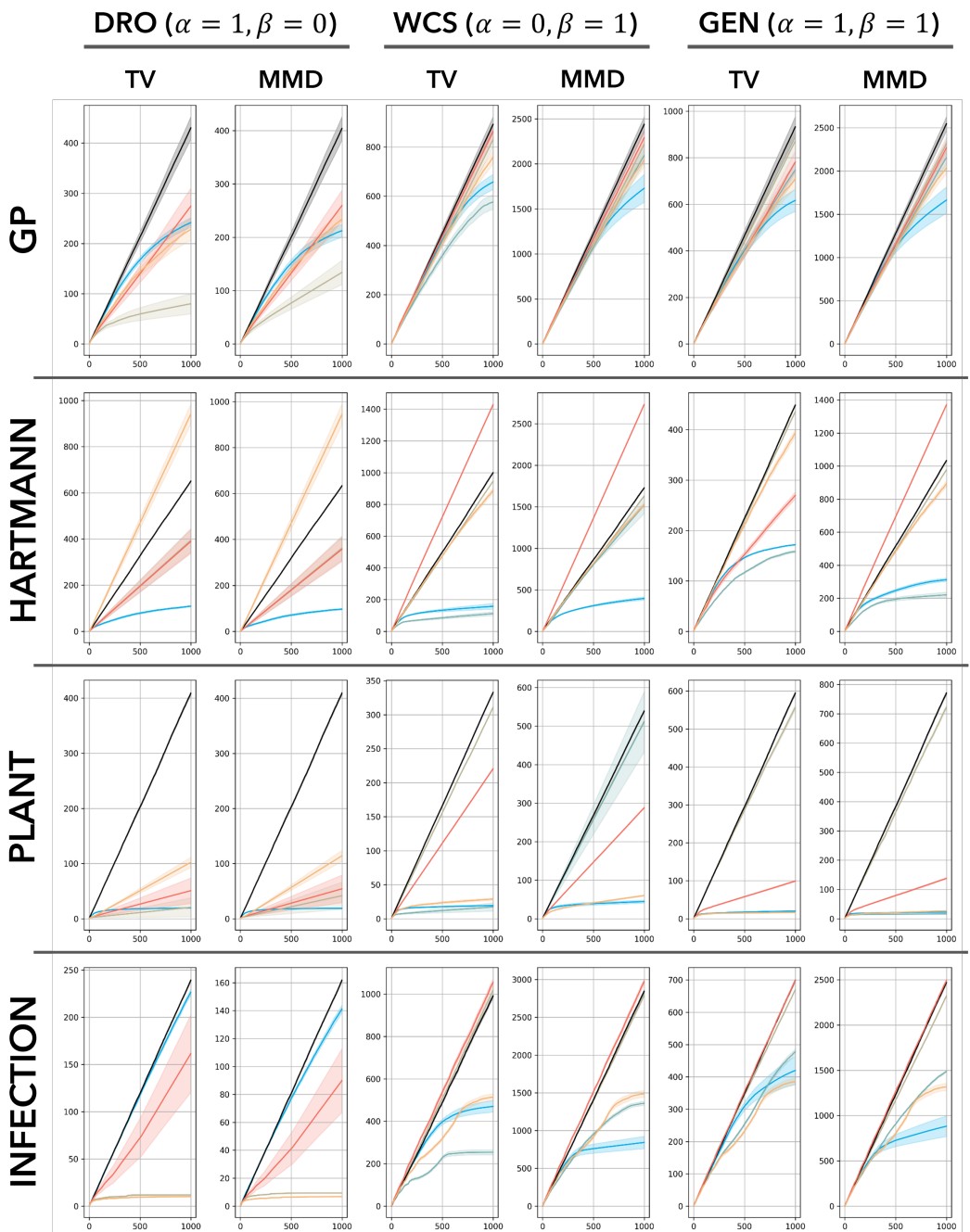

Figure 2: Mean and one standard error (over 10 RNG seeds) of cumulative regret (lower is better) incurred against iterations by the acquisitions **Random**, **UCB-SO**, **UCB-RO**, **UCB-BOCU-1**, **UCB-BOCU-2**, and **TS-BOCU** with varying objective functions, uncertainty objectives, and distribution distances. **UCB-BOCU-2** does not appear in the DRO experiments as it is equivalent to **UCB-BOCU-1** when $\beta = 0$.

## 5 EXPERIMENTS

**Experimental settings.** We evaluate our algorithm with 4 different underlying functions $f$: 1) samples from a GP prior; 2) the Hartmann 3-D function; 3) a plant growth simulator constructed

from real-world data where the decision and context variables are the pH and concentration of $NH_3$ of the nutrient medium respectively, and the output is the maximum leaf area of a plant (Tay et al., 2022); and 4) a COVID-19 epidemic model from Frazier et al. (2022) where the decision variables are proportions of test kits allocated to 3 different sub-groups, the context variables are the initial number of cases within these sub-groups and the transmission probability, and the output is the number of resultant cases. The decision and context variables for the GP samples and Hartmann function are decided from among their input variables arbitrarily. We choose 3 special cases of our general uncertainty objective: 1) the DRO objective with $\alpha = 1, \beta = 0$; 2) the WCS objective with $\alpha = 0, \beta = 1$; and 3) a general objective (termed GEN) not studied in previous work with $\alpha = 1, \beta = 1$. We use TV and MMD as the distribution distances $d$. We use a Gaussian and a uniform distribution for the reference and true distributions respectively, and keep them constant for all iterations. For DRO and GEN, we set the margin $\epsilon$ to be the distance between the reference and true distributions; for WCS, $\epsilon = 0$ as per its definition. Refer to Appendix C.1 for a full description of the experimental settings.

**Baselines.** To the best of our knowledge, our algorithm is the only algorithm in the literature designed to handle all 3 uncertainty objectives tested; we are not aware of any BO works that handle the WCS (for TV and MMD) and GEN objectives within the BOCU learning procedure. Nevertheless, we empirically compare our algorithm's ability to solve general uncertainty objectives to that of previous algorithms (and naive extensions) in the BOCU literature. We include a simple random acquisition baseline, UCB algorithms for SO and RO as used in Kirschner et al. (2020), and two naive extensions of the UCB algorithm for DRBO from Kirschner et al. (2020), named UCB-BOCU-1 and UCB-BOCU-2, that incorporate tractable but theoretically unsupported derivative 'upper bound' terms. When $\beta = 0$, these algorithms revert to the original DRBO algorithm. Refer to Appendix C.2 for a precise description of UCB-BOCU-1 and UCB-BOCU-2.

**Analysis.** The cumulative regret plots for each experimental setting are shown in Fig. 2. We observe that TS-BOCU consistently demonstrates the sublinear property across all experimental settings and achieves a low cumulative regret relative to the baselines in most settings. Random and SO tend to struggle on most settings while RO has inconsistent performance. The UCB-BOCU variants perform well on the DRO setting as expected since they were designed for that setting. However, UCB-BOCU-1 does not perform well on any setting where $\beta = 1$ and the uncertainty objective includes the derivative term. UCB-BOCU-2 performs better in these settings, but uses a theoretically unsupported heuristic. TS-BOCU enjoys both theoretical guarantees and good empirical performance. Finally, we observe that TS-BOCU performs well when MMD is used even though the theory was proved only for TV, which lends credence to the belief that the theory can be extended to a more general class of distribution distances. The code for the experiments may be found at `https://github.com/sebtsh/unified-framework-BOCU`.

## 6 CONCLUSION

This work proposed the BOCU framework, a natural extension of DRBO that unifies several uncertainty objectives studied in the BO literature, and developed a general algorithm that optimizes any uncertainty objective within the framework and whose performance is both theoretically and empirically supported. We believe our work lays the foundations for several directions of future work: one interesting question is whether the assumptions of Theorem 4.1 can be relaxed, such as by considering a more general class of distribution distances. Furthermore, as described in the related work section, there are still many disparate BO lines of research such as BO with uncertain inputs (Oliveira et al., 2019) and causal BO Aglietti et al. (2020) that do not fit within the BOCU framework. It is an open question as to whether there are even more general yet still theoretically interesting frameworks that subsume these other settings.

## REPRODUCIBILITY STATEMENT

We follow standard practices to ensure the reproducibility of our work. The source code for the experiments (along with all datasets) is provided in the supplementary material (available online at `https://github.com/sebtsh/unified-framework-BOCU`) for full reproducibility of

the experimental results. All algorithms are fully described, and all theoretical results have clearly stated assumptions and full proofs in the appendix.

## ACKNOWLEDGEMENT

This research/project is supported by A*STAR under its RIE2020 Advanced Manufacturing and Engineering (AME) Industry Alignment Fund – Pre Positioning (IAF-PP) (Award A19E4a0101).

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

# A PROOFS

## A.1 PROOF OF PROPOSITION 3.1

**Proposition 3.1.** *Suppose the distribution distance $d$ is such that $\delta(\mathbf{x}, 0)$ is finite. Fix a $\tau \leq \min_{\mathbf{x} \in \mathcal{X}} \mathbb{E}_{\mathbf{c} \sim \mathbf{p}} [f(\mathbf{x}, \mathbf{c})]$. For all $\mathbf{x} \in \mathcal{X}$, there exists an $\epsilon_{\mathbf{x}} \geq 0$ such that*

$$g_{RS}(\mathbf{x}; \tau) \in \partial v(\mathbf{x}, \epsilon_{\mathbf{x}})$$

*where $\partial v(\mathbf{x}, \epsilon_{\mathbf{x}})$ is the subdifferential of $v(\mathbf{x}, \epsilon)$ at $\epsilon_{\mathbf{x}}$.*

*Conversely, fix an $\epsilon \geq 0$. For all $\mathbf{x} \in \mathcal{X}$, there exists a $\tau_{\mathbf{x}} \leq \mathbb{E}_{\mathbf{c} \sim \mathbf{p}} [f(\mathbf{x}, \mathbf{c})]$ such that*

$$\delta(\mathbf{x}, \epsilon) = g_{RS}(\mathbf{x}; \tau_{\mathbf{x}}) ;$$

*in particular,*

$$g_{WCS}(\mathbf{x}) = \delta(\mathbf{x}, 0) = g_{RS}\left(\mathbf{x}; \mathbb{E}_{\mathbf{c} \sim \mathbf{p}} [f(\mathbf{x}, \mathbf{c})]\right) . \tag{12}$$

*Proof.* Part 1:

For the first part of the proposition, the subdifferential $\partial v(\mathbf{x}, \epsilon_{\mathbf{x}})$ is defined as

$$\partial v(\mathbf{x}, \epsilon_{\mathbf{x}}) := \{c \in \mathbb{R} : \forall \epsilon' \geq 0, v(\mathbf{x}, \epsilon') \geq v(\mathbf{x}, \epsilon_{\mathbf{x}}) + c(\epsilon' - \epsilon_{\mathbf{x}})\} ,$$

and we will show that $g_{RS}(\mathbf{x}; \tau)$ is in this set for some $\epsilon_{\mathbf{x}}$.

Let $k = g_{RS}(\mathbf{x}; \tau)$, which is to say that $k$ is the largest non-positive real number such that the following inequality holds for all $\epsilon \geq 0$:

$$v(\mathbf{x}, \epsilon) \geq \tau + k\epsilon . \tag{13}$$

If $k = 0$, then $k$ is in $\partial v(\mathbf{x}, \hat{\epsilon}_{\mathbf{x}})$ since $\delta(\mathbf{x}, \hat{\epsilon}_{\mathbf{x}}) \in \partial v(\mathbf{x}, \hat{\epsilon}_{\mathbf{x}})$ and $\delta(\mathbf{x}, \hat{\epsilon}_{\mathbf{x}}) = 0$.

If $k < 0$, we first show that this inequality holds with equality for at least one value of $\epsilon$. As a proof by contradiction, suppose not, i.e., $v(\mathbf{x}, \epsilon) > \tau + k\epsilon$ for all $\epsilon \geq 0$. Then, the gap $b(\epsilon) := v(\mathbf{x}, \epsilon) - \tau - k\epsilon > 0$ for all $\epsilon \geq 0$.

$b(\epsilon)$ is a continuous function on the interval $[0, \hat{\epsilon}_{\mathbf{x}}]$ by the following argument: Since $v(\mathbf{x}, \epsilon)$ is a convex function of $\epsilon$ with domain $[0, \infty)$, it is continuous everywhere on the open interval $(0, \infty)$. Since the proposition assumes that $\delta(\mathbf{x}, 0)$ is finite, $v(\mathbf{x}, \epsilon)$ is right-differentiable at $\epsilon = 0$ and is thus right-continuous at $\epsilon = 0$. $v(\mathbf{x}, \epsilon)$ is thus a continuous function on the interval $[0, \hat{\epsilon}_{\mathbf{x}}]$, and thus so is $b(\epsilon)$.

Since $b(\epsilon)$ is continuous on $[0, \hat{\epsilon}_{\mathbf{x}}]$, by the extreme value theorem, $c := \min_{\epsilon \in [0, \hat{\epsilon}_{\mathbf{x}}]} b(\epsilon) > 0$ exists. $c$ is the smallest gap between $v(\mathbf{x}, \epsilon)$ and $\tau + k\epsilon$ on the interval $[0, \hat{\epsilon}_{\mathbf{x}}]$. Now choose

$$k' := k + \min\left(\frac{k}{2}, \frac{c}{\hat{\epsilon}_{\mathbf{x}}}\right) .$$

$k' > k$ and, since $k < 0$, $k' \leq k/2 < 0$. Furthermore, for all $\epsilon \in [0, \hat{\epsilon}_{\mathbf{x}}]$,

$$\begin{aligned}
\tau + k'\epsilon &= \tau + k\epsilon + \min\left(\frac{k}{2}, \frac{c}{\hat{\epsilon}_{\mathbf{x}}}\right)\epsilon \\
&\leq \tau + k\epsilon + c\frac{\epsilon}{\hat{\epsilon}_{\mathbf{x}}} \\
&\leq \tau + k\epsilon + c \\
&\leq v(\mathbf{x}, \epsilon) .
\end{aligned}$$

Note that, for $\epsilon > \hat{\epsilon}_{\mathbf{x}}$, $v(\mathbf{x}, \epsilon) = v(\mathbf{x}, \hat{\epsilon}_{\mathbf{x}})$, and $\tau + k'\epsilon$ is decreasing in $\epsilon$. We therefore conclude that $k'$ satisfies the required inequality constraint Equation 13 for all $\epsilon > 0$, which leads to a contradiction since $0 < k' < k$ but $k$ was defined to be the largest valid number. We therefore conclude that Equation 13 holds with equality for at least one value of $\epsilon$.

Let $\epsilon_{\mathbf{x}}$ be this value of $\epsilon$ that satisfies Equation 13 with equality. For all $\epsilon \geq 0$, $\tau + k\epsilon$ can be rewritten as

$$\tau + k\epsilon = v(\mathbf{x}, \epsilon_{\mathbf{x}}) + k(\epsilon - \epsilon_{\mathbf{x}})$$

Since $\tau + k\epsilon \leq v(\mathbf{x}, \epsilon)$ for all $\epsilon \geq 0$, we conclude that $k$ is in $\partial v(\mathbf{x}, \epsilon_{\mathbf{x}})$ which completes the proof of the first part of the proposition.

Part 2:

For the second part of the proposition, with a fixed $\epsilon \geq 0$, define the linear function of $\epsilon' \geq 0$

$$a(\epsilon') := v(\mathbf{x}, \epsilon) + \delta(\mathbf{x}, \epsilon)(\epsilon' - \epsilon) .$$

By the convexity of $v(\mathbf{x}, \epsilon)$, $a(\epsilon') \leq v(\mathbf{x}, \epsilon')$ for all $\epsilon' \geq 0$. In particular,

$$\tau_{\mathbf{x}} := a(0) \leq v(\mathbf{x}, 0) = \mathop{\mathbb{E}}_{\mathbf{c} \sim \mathbf{p}} [f(\mathbf{x}, \mathbf{c})]$$

as desired. Observe that $a(\epsilon')$ can be rewritten in terms of $\tau_{\mathbf{x}}$:

$$a(\epsilon') = \tau_{\mathbf{x}} + \delta(\mathbf{x}, \epsilon)\epsilon' .$$

To show that $\delta(\mathbf{x}, \epsilon) = g_{\text{RS}}(\mathbf{x}; \tau_{\mathbf{x}})$, it remains to show that there is no $0 \geq k > \delta(\mathbf{x}, \epsilon)$ such that $v(\mathbf{x}, \epsilon') \geq \tau_{\mathbf{x}} + k\epsilon'$ for all $\epsilon' \geq 0$. Let $k = \delta(\mathbf{x}, \epsilon) + \gamma$ for some $\gamma > 0$. If $\epsilon > 0$,

$$\begin{aligned}
\tau_{\mathbf{x}} + k\epsilon &= \tau_{\mathbf{x}} + (\delta(\mathbf{x}, \epsilon) + \gamma)\epsilon \\
&= a(\epsilon) + \gamma\epsilon \\
&= v(\mathbf{x}, \epsilon) + \gamma\epsilon \\
&> v(\mathbf{x}, \epsilon)
\end{aligned}$$

as desired. If $\epsilon = 0$, first observe that, by definition of $\delta(\mathbf{x}, 0)$,

$$\frac{v(\mathbf{x}, \epsilon') - v(\mathbf{x}, 0)}{\epsilon'} \to \delta(\mathbf{x}, 0) \text{ as } \epsilon' \downarrow 0.$$

Thus, there must be some $\epsilon' > 0$ such that

$$\frac{v(\mathbf{x}, \epsilon') - v(\mathbf{x}, 0)}{\epsilon'} < \delta(\mathbf{x}, 0) + \gamma .$$

Rearranging,

$$\begin{aligned}
v(\mathbf{x}, \epsilon') - v(\mathbf{x}, 0) &< (\delta(\mathbf{x}, 0) + \gamma)\epsilon' \\
v(\mathbf{x}, \epsilon') &< v(\mathbf{x}, 0) + (\delta(\mathbf{x}, 0) + \gamma)\epsilon' \\
v(\mathbf{x}, \epsilon') &< \tau_{\mathbf{x}} + k\epsilon'
\end{aligned}$$

as desired. To show Equation 12, choose $\epsilon = 0$ to obtain $\tau_{\mathbf{x}} = v(\mathbf{x}, 0) = \mathbb{E}_{\mathbf{c} \sim \mathbf{p}} [f(\mathbf{x}, \mathbf{c})]$, which completes the proof. □

## A.2 PROOF OF THEOREM 4.1

**Theorem 4.1.** *If the following assumptions hold:*

1. *The kernel $k$ is the squared exponential kernel or a Matérn kernel with $\nu > 2$;*

2. *the distribution distance $d$ is the total variation (TV) distance;*

3. *for all $t \leq T$, the true distribution $\mathbf{p}_t^*$ satisfies the following:*

    (a) *$d(\mathbf{p}_t, \mathbf{p}_t^*) \leq \epsilon_t$;*
    (b) *for all $c \in [|\mathcal{C}|]$, $p_{t,c}^* \geq p_{min} > 0$,*

*then TS-BOCU (Algorithm 1) incurs a $T$-period Bayesian regret bounded by*

$$\text{BayesRegret}(T) \leq \mathcal{O}\left( \alpha\left( \sqrt{T(A_1\gamma_T + A_2) \ln (T^2\tau_t^m|\mathcal{C}|)} + \sum_{t=1}^{T} (\mathbb{E}[B\epsilon_t]) \right) \right.$$

$$\left. + \beta\left( \sqrt{\ln (T^2\tau_t^m|\mathcal{C}|)} + \sqrt{\ln |\mathcal{C}|} + p_{min}^{-1}\sqrt{T(A_1\gamma_T + A_2) \ln (T^2\tau_t^m|\mathcal{C}|)} \right) \right)$$

$$\tag{14}$$

*where $\gamma_T := \max_{\{(\mathbf{x}_t, \mathbf{c}_t)\}_{t=1}^{T}} (1/2) \ln |\mathbf{I} + \sigma^{-2}\mathbf{K}_T|$ is the maximum information gain, $B := \max_{\mathbf{x} \in \mathcal{X}} \|\mathbf{f}(\mathbf{x})\|$, $\tau_t := t^2 mab$, $A_1 := \ln(1 + \sigma^{-2})^{-1}$, $A_2 := \frac{1}{6}Lm^2ab\pi^{5/2}$, and $L$, $a$ and $b$ are kernel-dependent constants.*

*Proof.* Under the assumption that the kernel $k$ is the squared exponential kernel or a Matérn kernel with $\nu > 2$, GP sample paths $f$ satisfy the high probability bound $\mathbb{P}(\sup_{(\mathbf{x},\mathbf{c}) \in \mathcal{X} \times \mathcal{C}} |\frac{\partial f(\mathbf{x},\mathbf{c})}{\partial x_i}| > J) \leq ae^{-(J/b)^2}$ for all $i \in \{1, ..., m\}$ for some kernel-dependent constants $a$ and $b$ (Ghosal & Roy, 2006).

To enable the analysis of a continuous, infinite decision set $\mathcal{X} \subset \mathbb{R}^m$, we adopt the iterative discretization method first developed by Srinivas et al. (2010) and refined for Thompson sampling by Kandasamy et al. (2018). The general idea is to use an increasingly fine sequence of discretizations of $\mathcal{X}$ and show that, due to the assumed high probability bound on the GP sample paths, a decision $\mathbf{x}$ and its closest neighbour in the $t$-th discretization $[\mathbf{x}]_t$ share very similar function values.

Recall that $\mathcal{X}$ is a compact and convex subset of $\mathbb{R}^m$; for simplicity and without loss of generality, in this proof, we assume $\mathcal{X}$ is a subset of the $[0,1]^m$ hypercube. Let $\nu_t$ be the discretization of the $[0,1]^m$ hypercube at iteration $t$ with $\tau_t := t^2 mab\sqrt{\pi}$ equally spaced decisions along each dimension. Define

$$\tilde{\mathcal{X}}_t := \mathcal{X} \cap \nu_t \, ,$$

such that $|\tilde{\mathcal{X}}_t| \leq \tau_t^m$. $[\mathbf{x}]_t$ is then defined as $[\mathbf{x}]_t := \arg\min_{\mathbf{x}' \in \tilde{\mathcal{X}}_t} \|\mathbf{x} - \mathbf{x}'\|_2$. Note that $\|\mathbf{x} - [\mathbf{x}]_t\|_2 \leq \|\mathbf{x} - [\mathbf{x}]_t\|_1 \leq m/\tau_t$ for all $\mathbf{x} \in \mathcal{X}$.

In this proof, to reduce clutter, we define (making the dependence on $\epsilon_t, f, \mathbf{p}_t, d$ explicit)

$$v_t(\mathbf{x}) := v(\mathbf{x}, \epsilon_t; f, \mathbf{p}_t, d)$$
$$\delta_t(\mathbf{x}) := \delta(\mathbf{x}, \epsilon_t; f, \mathbf{p}_t, d) \, .$$

The proof relies on an upper confidence bound sequence $U_t$ of the form

$$U_t(\mathbf{x}) := \alpha \hat{v}_t(\mathbf{x}) + \beta \hat{\delta}_t(\mathbf{x})$$
$$\hat{v}_t(\mathbf{x}) := \inf_{\mathbf{q} \in \mathcal{U}(\mathbf{p}_t, d, \epsilon_t)} \mathbf{q}^\top \mathbf{u}_t(\mathbf{x})$$
$$\hat{\delta}_t(\mathbf{x}) := \max_{\boldsymbol{\ell}_t(\mathbf{x}) \preceq \boldsymbol{\varphi} \preceq \mathbf{u}_t(\mathbf{x})} \lim_{h \to 0} \frac{1}{h} \left( \inf_{\mathbf{q} \in \mathcal{U}(\mathbf{p}_t, d, \epsilon_t + h)} \mathbf{q}^\top \boldsymbol{\varphi} - \inf_{\mathbf{q} \in \mathcal{U}(\mathbf{p}_t, d, \epsilon_t)} \mathbf{q}^\top \boldsymbol{\varphi} \right)$$
$$\mathbf{u}_t(\mathbf{x}) := \boldsymbol{\mu}_{t-1}(\mathbf{x}) + \sqrt{\beta_t} \boldsymbol{\sigma}_{t-1}(\mathbf{x}), \quad \boldsymbol{\ell}_t(\mathbf{x}) := \boldsymbol{\mu}_{t-1}(\mathbf{x}) - \sqrt{\beta_t} \boldsymbol{\sigma}_{t-1}(\mathbf{x})$$

where $\{\beta_t\}_{t=1}^T$ is a sequence defined as $\beta_t = 2 \ln((t^2 + 1)|\tilde{\mathcal{X}}_t||\mathcal{C}|(2\pi)^{-1/2})$.

The initial part of the proof adapts the method of Russo & Van Roy (2014) and Kandasamy et al. (2018) in expressing the $T$-period Bayes regret of a Thompson sampling algorithm (accounting for discretization) as:

$$\text{BayesRegret}(T) = \sum_{t=1}^T \mathbb{E}\left[g_t(\mathbf{x}_t^*) - g_t(\mathbf{x}_t)\right]$$

$$= \sum_{t=1}^T \mathbb{E}\left[\mathbb{E}\left[g_t(\mathbf{x}_t^*) - g_t(\mathbf{x}_t) \,|\, H_t\right]\right]$$

$$\stackrel{(i)}{=} \sum_{t=1}^T \mathbb{E}[\mathbb{E}[g_t(\mathbf{x}_t^*) - g_t(\mathbf{x}_t) + U_t([\mathbf{x}_t]_t) - U_t([\mathbf{x}_t^*]_t) + g_t([\mathbf{x}_t^*]_t) - g_t([\mathbf{x}_t^*]_t)$$
$$+ g_t([\mathbf{x}_t]_t) - g_t([\mathbf{x}_t]_t) \,|\, H_t]]$$

$$= \sum_{t=1}^T \mathbb{E}\Big[\left(U_t([\mathbf{x}_t]_t) - g_t([\mathbf{x}_t]_t)\right) + \left(g_t([\mathbf{x}_t^*]_t - U_t([\mathbf{x}_t^*]_t)\right) + \left(g_t([\mathbf{x}_t]_t) - g_t(\mathbf{x}_t)\right)$$
$$+ \left(g_t(\mathbf{x}_t^*) - g_t([\mathbf{x}_t^*]_t)\right)\Big]$$

$$= \sum_{t=1}^T \mathbb{E}\Big[\alpha(\hat{v}_t([\mathbf{x}_t]_t) - v_t([\mathbf{x}_t]_t)) + \beta(\hat{\delta}_t([\mathbf{x}_t]_t) - \delta_t([\mathbf{x}_t]_t))$$
$$+ \alpha(v_t([\mathbf{x}_t^*]_t) - \hat{v}_t([\mathbf{x}_t^*]_t)) + \beta(\delta_t([\mathbf{x}_t^*]_t) - \hat{\delta}_t([\mathbf{x}_t^*]_t))$$
$$+ \alpha(v_t([\mathbf{x}_t]_t) - v_t(\mathbf{x}_t)) + \beta(\delta_t([\mathbf{x}_t]_t) - \delta_t(\mathbf{x}_t))$$

$$+ \alpha(v_t([\mathbf{x}_t^*]_t) - v_t(\mathbf{x}_t^*)) + \beta(\delta_t([\mathbf{x}_t^*]_t) - \delta_t(\mathbf{x}_t^*))\Big]$$

$$= \sum_{t=1}^{T} \alpha \underbrace{\left(\mathbb{E}\left[\hat{v}_t([\mathbf{x}_t]_t) - v_t([\mathbf{x}_t]_t)\right] + \mathbb{E}\left[v_t([\mathbf{x}_t^*]_t) - \hat{v}_t([\mathbf{x}_t^*]_t)\right]\right)}_{D_1}$$

$$+ \beta \underbrace{\left(\mathbb{E}\left[\hat{\delta}_t([\mathbf{x}_t]_t) - \delta_t([\mathbf{x}_t]_t)\right] + \mathbb{E}\left[\delta_t([\mathbf{x}_t^*]_t) - \hat{\delta}_t([\mathbf{x}_t^*]_t)\right]\right)}_{D_2}$$

$$+ \alpha \underbrace{\left(\mathbb{E}\left[v_t([\mathbf{x}_t]_t) - v_t(\mathbf{x}_t)\right] + \mathbb{E}\left[v_t([\mathbf{x}_t^*]_t) - v_t(\mathbf{x}_t^*)\right]\right)}_{D_3}$$

$$+ \beta \underbrace{\left(\mathbb{E}\left[\delta_t([\mathbf{x}_t]_t) - \delta_t(\mathbf{x}_t)\right] + \mathbb{E}\left[\delta_t([\mathbf{x}_t^*]_t) - \delta_t(\mathbf{x}_t^*)\right]\right)}_{D_4}$$

where $(i)$ follows since, conditioned on the history $H_t$, $U_t$ is deterministic and $\mathbf{x}_t$ and $\mathbf{x}_t^*$ are identically distributed due to the manner in which each $\mathbf{x}_t$ is chosen (Thompson sampling), therefore $[\mathbf{x}_t]$ and $[\mathbf{x}_t^*]$ are identically distributed as well.

Applications of Lemma A.1 to bound $D_1$ (DRO objective regret), Lemma A.3 to bound $D_2$ (DRO derivative regret), Lemma A.6 to bound $D_3$ (DRO objective discretization error), and Lemma A.7 to bound $D_4$ (DRO derivative discretization error) bound the terms in the sum separately to arrive at

$$\text{BayesRegret}(T) \leq \alpha \mathcal{O}\left(\sqrt{T(A_1\gamma_T + A_2)\ln\left(T^2|\tilde{\mathcal{X}}_t||\mathcal{C}|\right)} + \sum_{t=1}^{T} \mathbb{E}\left[B\epsilon_t\right]\right)$$

$$+ \beta \mathcal{O}\left(p_{\min}^{-1}\sqrt{T(A_1\gamma_T + A_2)\ln\left(T^2|\tilde{\mathcal{X}}_t||\mathcal{C}|\right)} + \sqrt{\ln\left(T^2|\tilde{\mathcal{X}}_t||\mathcal{C}|\right)} + \sqrt{\ln|\mathcal{C}|}\right)$$

$$+ \alpha\left(\frac{\pi^2}{6} + \sum_{t=1}^{T} 4\mathbb{E}\left[B\epsilon_t\right]\right) + \frac{\beta\pi^2}{3}$$

$$\leq \mathcal{O}\left(\alpha\left(\sqrt{T(A_1\gamma_T + A_2)\ln\left(T^2\tau_t^m|\mathcal{C}|\right)} + \sum_{t=1}^{T}\left(\mathbb{E}\left[B\epsilon_t\right]\right)\right)\right.$$

$$\left. + \beta\left(\sqrt{\ln\left(T^2\tau_t^m|\mathcal{C}|\right)} + \sqrt{\ln|\mathcal{C}|} + p_{\min}^{-1}\sqrt{T(A_1\gamma_T + A_2)\ln\left(T^2\tau_t^m|\mathcal{C}|\right)}\right)\right)$$

which completes the proof. $\qquad\square$

### A.3   PROOF OF LEMMA A.1

**Lemma A.1.** *If the following assumptions hold:*

1.  *The kernel $k$ is the squared exponential kernel or a Matérn kernel with $\nu > 2$;*

2.  *for all $t \leq T$, the true distribution $\mathbf{p}_t^*$ satisfies $d(\mathbf{p}_t, \mathbf{p}_t^*) \leq \epsilon_t$ for all $t \leq T$,*

*then the sequence of decisions $\{\mathbf{x}_t\}_{t=1}^{T}$ chosen by Algorithm 1 has a corresponding sequence of discretized decisions $\{[\mathbf{x}_t]_t\}_{t=1}^{T}$ that satisfies*

$$\sum_{t=1}^{T} \mathbb{E}\left[\hat{v}_t([\mathbf{x}_t]_t) - v_t([\mathbf{x}_t]_t)\right] + \mathbb{E}\left[v_t([\mathbf{x}_t^*]_t) - \hat{v}_t([\mathbf{x}_t^*]_t)\right] \leq \mathcal{O}\left(\sqrt{T(A_1\gamma_T + A_2)\ln\left(T^2|\tilde{\mathcal{X}}_t||\mathcal{C}|\right)}\right.$$

$$\left. + \sum_{t=1}^{T} \mathbb{E}\left[B\epsilon_t\right]\right) \qquad (15)$$

*where $v_t(\mathbf{x}) := v(\mathbf{x}, \epsilon_t; f, \mathbf{p}_t, d)$, $A_1 := \ln(1+\sigma^{-2})^{-1}$, $A_2 := \frac{1}{6}Lm^2ab\pi^{5/2}$, $L$, $a$ and $b$ are kernel-dependent constants, and $B$ is a complexity parameter that depends on distribution distance $d$ and*

$f$. For example, $B$ has the following specific forms for maximum mean discrepancy (MMD) and total variation (TV):

$$B := \begin{cases} \max_{\mathbf{x} \in \mathcal{X}} \|\mathbf{f}(\mathbf{x})\|_{M^{-1}} & \text{if } d = \text{MMD} , \\ \max_{\mathbf{x} \in \mathcal{X}} \|\mathbf{f}(\mathbf{x})\| & \text{if } d = \text{TV} . \end{cases}$$

*Proof.* Define

$$\mathbf{q}_t^u := \underset{\mathbf{q} \in \mathcal{U}(\mathbf{p}_t, d, \epsilon_t)}{\arg \min} \mathbf{q}^\top \mathbf{u}_t([\mathbf{x}_t]_t)$$

$$\mathbf{q}_t^f := \underset{\mathbf{q} \in \mathcal{U}(\mathbf{p}_t, d, \epsilon_t)}{\arg \min} \mathbf{q}^\top \mathbf{f}([\mathbf{x}_t]_t)$$

We first bound the sum of $\mathbb{E}\left[\hat{v}_t([\mathbf{x}_t]_t) - v_t([\mathbf{x}_t]_t)\right]$ terms in Equation 15. The proof adapts the proof techniques of of Kirschner et al. (2020) to the Thompson sampling algorithm.

$$\sum_{t=1}^T \mathbb{E}\left[\hat{v}_t([\mathbf{x}_t]_t) - v_t([\mathbf{x}_t]_t)\right] = \sum_{t=1}^T \mathbb{E}\left[\mathbf{q}_t^{u\top} \mathbf{u}_t([\mathbf{x}_t]_t) - \mathbf{q}_t^{f\top} \mathbf{f}([\mathbf{x}_t]_t)\right]$$

$$\leq \sum_{t=1}^T \mathbb{E}\left[\mathbf{p}_t^{*\top} \mathbf{u}_t([\mathbf{x}_t]_t) - \mathbf{q}_t^{f\top} \mathbf{f}([\mathbf{x}_t]_t)\right]$$

$$= \sum_{t=1}^T \mathbb{E}\left[\mathbf{p}_t^{*\top}(\mathbf{u}_t([\mathbf{x}_t]_t) - \mathbf{f}([\mathbf{x}_t]_t)) + (\mathbf{p}_t^* - \mathbf{q}_t^f)^\top \mathbf{f}([\mathbf{x}_t]_t)\right] \quad (16)$$

We first focus on each $\mathbb{E}\left[\mathbf{p}_t^{*\top}(\mathbf{u}_t([\mathbf{x}_t]_t) - \mathbf{f}([\mathbf{x}_t]_t))\right]$ term. The following analysis repeatedly applies the law of total expectation and makes explicit the random variables with which the expectations are taken with respect to (recall that all expectations without explicit subscripts are taken with respect to $f$ and $H_T$).

$$\mathbb{E}\left[\mathbf{p}_t^{*\top}(\mathbf{u}_t([\mathbf{x}_t]_t) - \mathbf{f}([\mathbf{x}_t]_t))\right] \overset{(i)}{=} \underset{f, H_t \setminus \{\mathbf{c}_t\}}{\mathbb{E}} \left[\mathbf{p}_t^{*\top}(\mathbf{u}_t([\mathbf{x}_t]_t) - \mathbf{f}([\mathbf{x}_t]_t))\right] \quad (17)$$

$$\overset{(ii)}{=} \underset{f, H_t \setminus \{\mathbf{c}_t\}}{\mathbb{E}} \left[\underset{\mathbf{c}_t}{\mathbb{E}}\left[u_t([\mathbf{x}_t]_t, \mathbf{c}_t) - f([\mathbf{x}_t]_t, \mathbf{c}_t) \mid f, H_t \setminus \{\mathbf{c}_t\}\right]\right]$$

$$= \underset{f, H_t}{\mathbb{E}} \left[u_t([\mathbf{x}_t]_t, \mathbf{c}_t) - f([\mathbf{x}_t]_t, \mathbf{c}_t)\right]$$

$$= \underset{H_t}{\mathbb{E}} \left[\underset{f}{\mathbb{E}}\left[u_t([\mathbf{x}_t]_t, \mathbf{c}_t) - f([\mathbf{x}_t]_t, \mathbf{c}_t) \mid H_t\right]\right]$$

$$\overset{(iii)}{=} \underset{H_t}{\mathbb{E}} \left[u_t([\mathbf{x}_t]_t, \mathbf{c}_t) - \underset{f}{\mathbb{E}}\left[f([\mathbf{x}_t]_t, \mathbf{c}_t) \mid H_t\right]\right]$$

$$= \underset{H_t}{\mathbb{E}} \left[u_t([\mathbf{x}_t]_t, \mathbf{c}_t) - \mu_{t-1}([\mathbf{x}_t]_t, \mathbf{c}_t)\right]$$

$$= \underset{H_t}{\mathbb{E}} \left[\sqrt{\beta_t} \sigma_{t-1}([\mathbf{x}_t]_t, \mathbf{c}_t)\right]$$

$$= \mathbb{E}\left[\sqrt{\beta_t} \sigma_{t-1}([\mathbf{x}_t]_t, \mathbf{c}_t)\right] \quad (18)$$

where $(i)$ follows since $\mathbf{u}_t([\mathbf{x}_t]_t)$ is a constant given all random variables in $H_t$ except $\mathbf{c}_t$, and $\mathbf{f}([\mathbf{x}_t]_t)$ is a constant given $f$ and $\mathbf{x}_t$; $(ii)$ follows since $\mathbf{c}_t \sim \mathbf{p}_t^*$ and $\mathbf{c}_t$ is independent of $f$ and $H_t \setminus \{\mathbf{c}_t\}$; and $(iii)$ follows since $u_t$ is a constant given $H_t$.

We refer to the proof of Theorem 4 in Tay et al. (2022) (Equation 81 onwards) to obtain

$$\mathbb{E}\left[(\mathbf{p}_t^* - \mathbf{q}_t^f)^\top \mathbf{f}([\mathbf{x}_t]_t)\right] \leq 2\mathbb{E}\left[B\epsilon_t\right]$$

$$B := \begin{cases} \max_{\mathbf{x} \in \mathcal{X}} \|\mathbf{f}(\mathbf{x})\|_{M^{-1}} & \text{if } d = MMD , \\ \max_{\mathbf{x} \in \mathcal{X}} \|\mathbf{f}(\mathbf{x})\| & \text{if } d = TV . \end{cases}$$

Substituting these into Equation 16, following Srinivas et al. (2010) and accounting for the discretization,

$$\sum_{t=1}^{T} \mathbb{E}\left[\hat{v}_t(\mathbf{x}_t) - v_t(\mathbf{x}_t)\right] \tag{19}$$

$$\leq \sum_{t=1}^{T} \mathbb{E}\left[\sqrt{\beta_t}\sigma_{t-1}([\mathbf{x}_t]_t, \mathbf{c}_t)\right] + 2\mathbb{E}\left[B\epsilon_t\right] \tag{20}$$

$$\overset{(i)}{\leq} \mathbb{E}\left[\sqrt{T\beta_T}\sqrt{\sum_{t=1}^{T}\sigma_{t-1}^2([\mathbf{x}_t]_t, \mathbf{c}_t)}\right] + 2\sum_{t=1}^{T}\mathbb{E}\left[B\epsilon_t\right]$$

$$\overset{(ii)}{\leq} \mathbb{E}\left[\sqrt{T\beta_T}\sqrt{\sum_{t=1}^{T}(\sigma_{t-1}^2(\mathbf{x}_t, \mathbf{c}_t) + \frac{4Lm}{\tau_t})}\right] + 2\sum_{t=1}^{T}\mathbb{E}\left[B\epsilon_t\right]$$

$$= \mathbb{E}\left[\sqrt{T\beta_T}\sqrt{\frac{2}{3}Lm^2ab\pi^{5/2} + \sigma^2\sum_{t=1}^{T}\sigma^{-2}\sigma_{t-1}^2(\mathbf{x}_t, \mathbf{c}_t)}\right] + 2\sum_{t=1}^{T}\mathbb{E}\left[B\epsilon_t\right]$$

$$\overset{(iii)}{\leq} \mathbb{E}\left[\sqrt{T\beta_T}\sqrt{\frac{2}{3}Lm^2ab\pi^{5/2} + \ln(1+\sigma^{-2})^{-1}\sum_{t=1}^{T}\ln(1+\sigma^{-2}\sigma_{t-1}^2(\mathbf{x}_t, \mathbf{c}_t))}\right] + 2\sum_{t=1}^{T}\mathbb{E}\left[B\epsilon_t\right]$$

$$\overset{(iv)}{\leq} \sqrt{T\beta_T\left(\gamma_T\ln(1+\sigma^{-2})^{-1} + \frac{2}{3}Lm^2ab\pi^{5/2}\right)} + 2\sum_{t=1}^{T}\mathbb{E}\left[B\epsilon_t\right] \tag{21}$$

where $(i)$ uses the Cauchy-Schwarz inequality; $(ii)$ follows from Lemma A.9 for some kernel-dependent constant $L$; $(iii)$ follows since $x/\ln(1+x) \geq 1$ and is increasing for all $x > 0$ and $k(\mathbf{x}, \mathbf{c}; \mathbf{x}, \mathbf{c}) = 1$, therefore $\sigma^{-2}\sigma_{t-1}^2(\mathbf{x}_t, \mathbf{c}_t) \leq \sigma^{-2}\ln(1+\sigma^{-2})^{-1}\ln(1+\sigma^{-2}\sigma_{t-1}^2(\mathbf{x}_t, \mathbf{c}_t))$; and $(iv)$ follows from Lemma 5.3 in Srinivas et al. (2010) on bounding the sum of predictive variances with the information gain of the selected points.

**Remark A.2.** *Step $(ii)$ of the above chain of inequalities is a necessary step that was omitted from the proof of Theorem 11 in Kandasamy et al. (2018), in Equation 8 of their paper. The sum of posterior variances of the discretized decisions $\{[\mathbf{x}_t]_t\}_{t=1}^{T}$ cannot be directly bound by the information gain, since each $[\mathbf{x}_t]_t$ was not selected by the algorithm and are present only in the analysis. The algorithm selects $\{\mathbf{x}_t\}_{t=1}^{T}$, and thus Lemma A.9 is needed to bound the difference in posterior variances.*

We next bound the sum of $\mathbb{E}\left[v_t([\mathbf{x}_t^*]_t) - \hat{v}_t([\mathbf{x}_t^*]_t)\right]$ terms in Equation 15 by adapting the proof of Lemma 2 in Russo & Van Roy (2014). For any decision $\mathbf{x} \in \mathcal{X}$, re-define

$$\mathbf{q}_t^u := \operatorname*{arg\,min}_{\mathbf{q}\in\mathcal{U}(\mathbf{p}_t, d, \epsilon_t)} \mathbf{q}^{\top}\mathbf{u}_t(\mathbf{x})$$

$$\mathbf{q}_t^f := \operatorname*{arg\,min}_{\mathbf{q}\in\mathcal{U}(\mathbf{p}_t, d, \epsilon_t)} \mathbf{q}^{\top}\mathbf{f}(\mathbf{x})$$

Considering the term at iteration $t$ for any such $\mathbf{x}$,

$$\mathbb{E}\left[v_t(\mathbf{x}) - \hat{v}_t(\mathbf{x})\right] = \mathbb{E}\left[\mathbf{q}_t^{f\top}\mathbf{f}(\mathbf{x}) - \mathbf{q}_t^{u\top}\mathbf{u}_t(\mathbf{x})\right]$$

$$\leq \mathbb{E}\left[\mathbf{q}_t^{u\top}(\mathbf{f}(\mathbf{x}) - \mathbf{u}_t(\mathbf{x}))\right]$$

$$= \mathbb{E}\left[\sum_{\mathbf{c}\in\mathcal{C}} q_{t,\mathbf{c}}^u(f(\mathbf{x}, \mathbf{c}) - u_t(\mathbf{x}, \mathbf{c}))\right]$$

$$\leq \mathbb{E}\left[\sum_{\mathbf{c}\in\mathcal{C}} q_{t,\mathbf{c}}^u(f(\mathbf{x}, \mathbf{c}) - u_t(\mathbf{x}, \mathbf{c}))\mathbb{1}[f(\mathbf{x}, \mathbf{c}) - u_t(\mathbf{x}, \mathbf{c}) > 0]\right]$$

$$\leq \mathbb{E}\left[\sum_{\mathbf{c}\in\mathcal{C}}(f(\mathbf{x}, \mathbf{c}) - u_t(\mathbf{x}, \mathbf{c}))\mathbb{1}[f(\mathbf{x}, \mathbf{c}) - u_t(\mathbf{x}, \mathbf{c}) > 0]\right]$$

$$= \sum_{\mathbf{c} \in \mathcal{C}} \mathbb{E} \left[ (f(\mathbf{x}, \mathbf{c}) - u_t(\mathbf{x}, \mathbf{c})) \mathbb{1}[f(\mathbf{x}, \mathbf{c}) - u_t(\mathbf{x}, \mathbf{c}) > 0]] \right]$$

$$= \sum_{\mathbf{c} \in \mathcal{C}} \mathbb{E}_{H_t} \left[ \mathbb{E} \left[ (f(\mathbf{x}, \mathbf{c}) - u_t(\mathbf{x}, \mathbf{c})) \mathbb{1}[f(\mathbf{x}, \mathbf{c}) - u_t(\mathbf{x}, \mathbf{c}) > 0] \mid H_t \right] \right]$$

Conditioned on $H_t$, $(f(\mathbf{x}, \mathbf{c}) - u_t(\mathbf{x}, \mathbf{c})) \sim \mathcal{N}(-\sqrt{\beta_t} \sigma_{t-1}(\mathbf{x}, \mathbf{c}), \sigma_{t-1}^2(\mathbf{x}, \mathbf{c}))$. Using the fact that, if $X \sim \mathcal{N}(\mu, \sigma^2)$ and $\mu \leq 0$, the expectation $\mathbb{E}[X \mathbb{1}[X > 0]] = \int_0^\infty \frac{x}{\sigma \sqrt{2\pi}} \exp(\frac{-(x-\mu)^2}{2\sigma^2}) \, dx \leq \frac{\sigma}{\sqrt{2\pi}} \exp(\frac{-\mu^2}{2\sigma^2})$, we continue with

$$\mathbb{E}[v_t(\mathbf{x}) - \hat{v}_t(\mathbf{x})] \leq \sum_{\mathbf{c} \in \mathcal{C}} \mathbb{E}_{H_t} \left[ \frac{\sigma_{t-1}(\mathbf{x}, \mathbf{c})}{\sqrt{2\pi}} \exp\left(-\frac{\beta_t}{2}\right) \right]$$

$$= \sum_{\mathbf{c} \in \mathcal{C}} \mathbb{E}_{H_t} \left[ \frac{\sigma_{t-1}(\mathbf{x}, \mathbf{c})}{(t^2 + 1)|\tilde{\mathcal{X}}_t||\mathcal{C}|} \right]$$

$$\leq \frac{1}{(t^2 + 1)|\tilde{\mathcal{X}}_t|}$$

where the last inequality follows since $k(\mathbf{x}, \mathbf{c}; \mathbf{x}, \mathbf{c}) = 1$ by assumption, for all $\mathbf{x} \in \mathcal{X}$ and $\mathbf{c} \in \mathcal{C}$. The sum of $\mathbb{E}[v_t([\mathbf{x}_t^*]_t) - \hat{v}_t([\mathbf{x}_t^*]_t)]$ terms can then be bounded with

$$\sum_{t=1}^T \mathbb{E}[v_t([\mathbf{x}_t^*]_t) - \hat{v}_t([\mathbf{x}_t^*]_t)] \leq \sum_{t=1}^T \sum_{\mathbf{x} \in \tilde{\mathcal{X}}_t} \max(0, \mathbb{E}[v_t(\mathbf{x}) - \hat{v}_t(\mathbf{x})])$$

$$\leq \sum_{t=1}^T \sum_{\mathbf{x} \in \tilde{\mathcal{X}}_t} \frac{1}{(t^2 + 1)|\tilde{\mathcal{X}}_t|}$$

$$\leq 1 .$$

Finally, combining this result with Equation 21 and substituting in the value of $\beta_T$, we arrive at the result

$$\sum_{t=1}^T \mathbb{E}[\hat{v}_t(\mathbf{x}_t) - v_t(\mathbf{x}_t)] + \mathbb{E}[v_t(\mathbf{x}_t^*) - \hat{v}_t(\mathbf{x}_t^*)]$$

$$\leq \sqrt{2T \left( \gamma_T \ln(1 + \sigma^{-2})^{-1} + \frac{2}{3} L m^2 a b \pi^{5/2} \right) \ln \left( T^2 |\tilde{\mathcal{X}}_t||\mathcal{C}|(2\pi)^{-1/2} \right)} + 2 \sum_{t=1}^T \mathbb{E}[B\epsilon_t] + 1$$

$$= \mathcal{O} \left( \sqrt{T(A_1 \gamma_T + A_2) \ln \left( T^2 |\tilde{\mathcal{X}}_t||\mathcal{C}| \right)} + \sum_{t=1}^T \mathbb{E}[B\epsilon_t] \right)$$

where $A_1 := \ln(1 + \sigma^{-2})^{-1}$ and $A_2 := \frac{2}{3} L m^2 a b \pi^{5/2}$, which completes the proof. $\qquad \square$

## A.4 PROOF OF LEMMA A.3

**Lemma A.3.** *If the assumptions of Theorem 4.1 hold, then the sequence of decisions $\{\mathbf{x}_t\}_{t=1}^T$ chosen by Algorithm 1 has a corresponding sequence of discretized decisions $\{[\mathbf{x}_t]_t\}_{t=1}^T$ that satisfies*

$$\sum_{t=1}^T \mathbb{E} \left[ \hat{\delta}_t([\mathbf{x}_t]_t) - \delta_t([\mathbf{x}_t]_t) \right] + \mathbb{E} \left[ \delta_t([\mathbf{x}_t^*]_t) - \hat{\delta}_t([\mathbf{x}_t^*]_t) \right]$$

$$\leq \mathcal{O} \left( p_{min}^{-1} \sqrt{T(A_1 \gamma_T + A_2) \ln \left( T^2 |\tilde{\mathcal{X}}_t||\mathcal{C}| \right)} + \sqrt{\ln \left( T^2 |\tilde{\mathcal{X}}_t||\mathcal{C}| \right)} + \sqrt{\ln |\mathcal{C}|} \right) \qquad (22)$$

*where $\delta_t(\mathbf{x}) := \delta(\mathbf{x}, \epsilon_t; f, \mathbf{p}_t, d)$, $A_1 := \ln(1 + \sigma^{-2})^{-1}$, $A_2 := \frac{1}{6} L m^2 a b \pi^{5/2}$, and $L$, $a$ and $b$ are kernel-dependent constants.*

*Proof.* Define the following:

$$\boldsymbol{\varphi}_t(\mathbf{x}) := \underset{\boldsymbol{\ell}_t(\mathbf{x}) \preceq \boldsymbol{\varphi} \preceq \mathbf{u}_t(\mathbf{x})}{\arg\max} \lim_{h \to 0} \frac{1}{h} \left( \min_{\mathbf{q} \in \mathcal{U}(\mathbf{p}_t, d, \epsilon_t + h)} \mathbf{q}^\top \boldsymbol{\varphi} - \min_{\mathbf{q} \in \mathcal{U}(\mathbf{p}_t, d, \epsilon_t)} \mathbf{q}^\top \boldsymbol{\varphi} \right)$$

$$\hat{\mathbf{q}}_t^h := \underset{\mathbf{q} \in \mathcal{U}(\mathbf{p}_t, d, \epsilon_t + h)}{\arg\min} \mathbf{q}^\top \boldsymbol{\varphi}_t([\mathbf{x}_t]_t)$$

$$\hat{\mathbf{q}}_t := \underset{\mathbf{q} \in \mathcal{U}(\mathbf{p}_t, d, \epsilon_t)}{\arg\min} \mathbf{q}^\top \boldsymbol{\varphi}_t([\mathbf{x}_t]_t)$$

$$\mathbf{q}_t^h := \underset{\mathbf{q} \in \mathcal{U}(\mathbf{p}_t, d, \epsilon_t + h)}{\arg\min} \mathbf{q}^\top \mathbf{f}([\mathbf{x}_t]_t)$$

$$\mathbf{q}_t := \underset{\mathbf{q} \in \mathcal{U}(\mathbf{p}_t, d, \epsilon_t)}{\arg\min} \mathbf{q}^\top \mathbf{f}([\mathbf{x}_t]_t)$$

We begin with each $\hat{\delta}_t([\mathbf{x}_t]_t) - \delta_t([\mathbf{x}_t]_t)$ term, leaving out the sum and expectation for now.

$$
\begin{aligned}
\hat{\delta}_t([\mathbf{x}_t]_t) - \delta_t([\mathbf{x}_t]_t) &= \lim_{h \to 0} \frac{1}{h} \left( (\hat{\mathbf{q}}_t^h - \hat{\mathbf{q}}_t)^\top \boldsymbol{\varphi}_t([\mathbf{x}_t]_t) - (\mathbf{q}_t^h - \mathbf{q}_t)^\top \mathbf{f}([\mathbf{x}_t]_t) \right) \\
&\overset{(i)}{=} \lim_{h \to 0} \frac{1}{h} \left( h(\mathbf{e}_{i^-} - \mathbf{e}_{i^+})^\top \boldsymbol{\varphi}_t([\mathbf{x}_t]_t) - h(\mathbf{e}_{j^-} - \mathbf{e}_{j^+})^\top \mathbf{f}([\mathbf{x}_t]_t) \right) \\
&= (\mathbf{e}_{i^-} - \mathbf{e}_{i^+})^\top \boldsymbol{\varphi}_t([\mathbf{x}_t]_t) - (\mathbf{e}_{j^-} - \mathbf{e}_{j^+})^\top \mathbf{f}([\mathbf{x}_t]_t) \\
&= \varphi_t([\mathbf{x}_t]_t)_{i^-} - \varphi_t([\mathbf{x}_t]_t)_{i^+} - f([\mathbf{x}_t]_t)_{j^-} + f([\mathbf{x}_t]_t)_{j^+} \\
&\leq \varphi_t([\mathbf{x}_t]_t)_{j^-} - f([\mathbf{x}_t]_t)_{j^-} + f([\mathbf{x}_t]_t)_{j^+} - \varphi_t([\mathbf{x}_t]_t)_{i^+} \\
&\overset{(ii)}{\leq} \varphi_t([\mathbf{x}_t]_t)_{j^-} - f([\mathbf{x}_t]_t)_{j^-} + f([\mathbf{x}_t]_t)_o - \varphi_t([\mathbf{x}_t]_t)_o \\
&\leq (u_t([\mathbf{x}_t]_t)_{j^-} - f([\mathbf{x}_t]_t)_{j^-}) + (f([\mathbf{x}_t]_t)_o - \ell_t([\mathbf{x}_t]_t)_o)
\end{aligned}
$$

where $(i)$ follows from applying Lemma A.12 and defining $i^- := \arg\min_i \varphi_t([\mathbf{x}_t]_t)_i$, $i^+ := \arg\max_i \{\varphi_t([\mathbf{x}_t]_t)_i : \hat{q}_{t,i} > 0\}$, and similarly defining $j^-$ and $j^+$ with respect to $\mathbf{f}([\mathbf{x}_t]_t)$ and $\mathbf{q}_t$; and $(ii)$ follows from applying Lemma A.13 where $o$ is some index in $[n]$.

Putting back the expectation, taking steps similar to those between Equation 17 and Equation 18 in the proof of Lemma A.1,

$$
\begin{aligned}
\mathbb{E}\left[ \hat{\delta}_t([\mathbf{x}_t]_t) - \delta_t([\mathbf{x}_t]_t) \right] &\leq \mathbb{E}\left[ (u_t([\mathbf{x}_t]_t)_{j^-} - f([\mathbf{x}_t]_t)_{j^-}) + (f([\mathbf{x}_t]_t)_o - \ell_t([\mathbf{x}_t]_t)_o) \right] \\
&= \underset{H_t \setminus \{\mathbf{c}_t\}}{\mathbb{E}} \left[ \left( u_t([\mathbf{x}_t]_t)_{j^-} - \underset{f}{\mathbb{E}}\left[ f([\mathbf{x}_t]_t)_{j^-} \mid H_t \setminus \{\mathbf{c}_t\} \right] \right) \right. \\
&\qquad \left. + \left( \underset{f}{\mathbb{E}}\left[ f([\mathbf{x}_t]_t)_o \mid H_t \setminus \{\mathbf{c}_t\} \right] - \ell_t([\mathbf{x}_t]_t)_o \right) \right] \\
&= \underset{H_t \setminus \{\mathbf{c}_t\}}{\mathbb{E}} \left[ (u_t([\mathbf{x}_t]_t)_{j^-} - \mu_{t-1}([\mathbf{x}_t]_t)_{j^-}) + (\mu_{t-1}([\mathbf{x}_t]_t)_o - \ell_t([\mathbf{x}_t]_t)_o) \right] \\
&= \underset{H_t \setminus \{\mathbf{c}_t\}}{\mathbb{E}} \left[ \sqrt{\beta_t} \sigma_{t-1}([\mathbf{x}_t]_t)_{j^-} + \sqrt{\beta_t} \sigma_{t-1}([\mathbf{x}_t]_t)_o \right] \\
&\leq 2 \underset{H_t \setminus \{\mathbf{c}_t\}}{\mathbb{E}} \sqrt{\beta_t} \left[ \max_{c \in [n]} \sigma_{t-1}([\mathbf{x}_t]_t)_c \right] \\
&\overset{(i)}{\leq} 2 \underset{H_t \setminus \{\mathbf{c}_t\}}{\mathbb{E}} \left[ \sqrt{\beta_t} \cdot \frac{1}{p_{\min}} \mathbf{p}_t^{*\top} \boldsymbol{\sigma}_{t-1}([\mathbf{x}_t]_t) \right] \\
&= \frac{2}{p_{\min}} \underset{H_t \setminus \{\mathbf{c}_t\}}{\mathbb{E}} \left[ \underset{\mathbf{c}_t}{\mathbb{E}} \left[ \sqrt{\beta_t} \sigma_{t-1}([\mathbf{x}_t]_t, \mathbf{c}_t) \mid H_t \setminus \{\mathbf{c}_t\} \right] \right] \\
&= \frac{2}{p_{\min}} \underset{H_t}{\mathbb{E}} \left[ \sqrt{\beta_t} \sigma_{t-1}([\mathbf{x}_t]_t, \mathbf{c}_t) \right] \\
&= \frac{2}{p_{\min}} \mathbb{E}\left[ \sqrt{\beta_t} \sigma_{t-1}([\mathbf{x}_t]_t, \mathbf{c}_t) \right] \qquad (23)
\end{aligned}
$$

where $(i)$ follows since $\max_{c \in [n]} \sigma_{t-1}([\mathbf{x}_t]_t)_c \leq \sum_{c=1}^n \sigma_{t-1}([\mathbf{x}_t]_t)_c \leq \frac{1}{p_{\min}} \mathbf{p}_t^{*\top} \boldsymbol{\sigma}_{t-1}([\mathbf{x}_t]_t)$, by definition of $p_{\min}$ in Assumption 4(a).

Observe that the right hand side of Equation 23 is that of Equation 18 in the proof of Lemma A.1 multiplied by a $\frac{2}{p_{\min}}$ constant. Summing over $t$ and following the steps from Equation 20 to Equation 21 in the proof of Lemma A.1 yields

$$\sum_{t=1}^T \mathbb{E}\left[\hat{\delta}_t(\mathbf{x}_t) - \delta_t(\mathbf{x}_t)\right] \leq \frac{2}{p_{\min}} \sqrt{T\beta_T \left(\gamma_T \ln(1 + \sigma^{-2})^{-1} + \frac{2}{3} Lm^2 ab\pi^{5/2}\right)}. \tag{24}$$

We now bound the sum of $\mathbb{E}\left[\delta_t(\mathbf{x}_t^*) - \hat{\delta}_t(\mathbf{x}_t^*)\right]$ terms in Equation 22. For any decision $\mathbf{x} \in \mathcal{X}$,

$$\mathbb{E}\left[(\delta_t(\mathbf{x}) - \hat{\delta}_t(\mathbf{x}))\mathbb{1}[\delta_t(\mathbf{x}) - \hat{\delta}_t(\mathbf{x}) > 0]\right] \overset{(i)}{\leq} \mathbb{E}\left[-\hat{\delta}_t(\mathbf{x})\mathbb{1}[\delta_t(\mathbf{x}) - \hat{\delta}_t(\mathbf{x}) > 0]\right]$$

$$= \underset{H_t}{\mathbb{E}}\left[\underset{f}{\mathbb{E}}\left[-\hat{\delta}_t(\mathbf{x})\mathbb{1}[\delta_t(\mathbf{x}) - \hat{\delta}_t(\mathbf{x}) > 0] \mid H_t\right]\right]$$

$$\overset{(ii)}{=} \underset{H_t}{\mathbb{E}}\left[(-\hat{\delta}_t(\mathbf{x}))\underset{f}{\mathbb{E}}\left[\mathbb{1}[\delta_t(\mathbf{x}) - \hat{\delta}_t(\mathbf{x}) > 0] \mid H_t\right]\right]$$

$$\overset{(iii)}{\leq} \underset{H_t}{\mathbb{E}}\left[(-\hat{\delta}_t(\mathbf{x}))\right]\left(\frac{\sqrt{2\pi}}{(t^2 + 1)|\tilde{\mathcal{X}}_t|}\right)$$

$$\overset{(iv)}{\leq} \frac{2\sqrt{2\pi}\left(\sqrt{\beta_t} + \sqrt{2\ln|\mathcal{C}|}\right)}{(t^2 + 1)|\tilde{\mathcal{X}}_t|}$$

where $(i)$ follows since $\delta_t(\mathbf{x}) \leq 0$; $(ii)$ follows since $\hat{\delta}_t(\mathbf{x})$ is a constant given $H_t \setminus \{\mathbf{c}_t\}$; $(iii)$ follows from Lemma A.4; and $(iv)$ follows from Lemma A.5.

The sum of $\mathbb{E}\left[\delta_t([\mathbf{x}_t^*]_t) - \hat{\delta}_t([\mathbf{x}_t^*]_t)\right]$ then becomes

$$\sum_{t=1}^T \mathbb{E}\left[\delta_t([\mathbf{x}_t^*]_t) - \hat{\delta}_t([\mathbf{x}_t^*]_t)\right] \leq \sum_{t=1}^T \mathbb{E}\left[(\delta_t([\mathbf{x}_t^*]_t) - \hat{\delta}_t([\mathbf{x}_t^*]_t))\mathbb{1}[\delta_t([\mathbf{x}_t^*]_t) - \hat{\delta}_t([\mathbf{x}_t^*]_t) > 0]\right]$$

$$\leq \sum_{t=1}^T \sum_{\mathbf{x} \in \tilde{\mathcal{X}}_t} \mathbb{E}\left[(\delta_t(\mathbf{x}) - \hat{\delta}_t(\mathbf{x}))\mathbb{1}[\delta_t(\mathbf{x}) - \hat{\delta}_t(\mathbf{x}) > 0]\right]$$

$$\leq \sum_{t=1}^T \sum_{\mathbf{x} \in \tilde{\mathcal{X}}_t} \frac{2\sqrt{2\pi}\left(\sqrt{\beta_t} + \sqrt{2\ln|\mathcal{C}|}\right)}{(t^2 + 1)|\tilde{\mathcal{X}}_t|}$$

$$\leq 2\sqrt{2\pi}\left(\sqrt{\beta_T} + \sqrt{2\ln|\mathcal{C}|}\right).$$

Combining this result with Equation 24 and substituting in the value of $\beta_T$ yields

$$\sum_{t=1}^T \mathbb{E}\left[\hat{\delta}_t([\mathbf{x}_t]_t) - \delta_t([\mathbf{x}_t]_t)\right] + \mathbb{E}\left[\delta_t([\mathbf{x}_t^*]_t) - \hat{\delta}_t([\mathbf{x}_t^*]_t)\right]$$

$$\leq \frac{2}{p_{\min}}\sqrt{T\beta_T\left(\gamma_T \ln(1 + \sigma^{-2})^{-1} + \frac{1}{6} Lm^2 ab\pi^{5/2}\right)} + 2\sqrt{2\pi}\left(\sqrt{\beta_T} + \sqrt{2\ln|\mathcal{C}|}\right)$$

$$= \mathcal{O}\left(p_{\min}^{-1}\sqrt{T(A_1\gamma_T + A_2)\ln\left(T^2|\tilde{\mathcal{X}}_t||\mathcal{C}|\right)} + \sqrt{\ln\left(T^2|\tilde{\mathcal{X}}_t||\mathcal{C}|\right)} + \sqrt{\ln|\mathcal{C}|}\right)$$

where $A_1 := \ln(1 + \sigma^{-2})^{-1}$ and $A_2 := \frac{1}{6} Lm^2 ab\pi^{5/2}$, which completes the proof. $\qquad\square$

## A.5 Proof of Lemma A.4

**Lemma A.4.** *Under the assumptions and definitions of Lemma A.3, for any* $\mathbf{x} \in \mathcal{X}$ *and* $t \leq T$,

$$\mathbb{E}_f\left[\mathbb{1}\left[\delta_t(\mathbf{x}) - \hat{\delta}_t(\mathbf{x}) > 0\right] \mid H_t\right] \leq \frac{\sqrt{2\pi}}{(t^2 + 1)|\tilde{\mathcal{X}}_t|} .$$

*Proof.* Let $A$ be the event $\{\delta_t(\mathbf{x}) - \hat{\delta}_t(\mathbf{x}) > 0\}$, and $B$ be the event $\{\boldsymbol{\ell}_t(\mathbf{x}) \preceq \mathbf{f}(\mathbf{x}) \preceq \mathbf{u}_t(\mathbf{x})\}^c$, i.e., the event that $\mathbf{f}(\mathbf{x})$ is outside the confidence bounds.

$$
\begin{aligned}
\mathbb{E}_f\left[\mathbb{1}\left[\delta_t(\mathbf{x}) - \hat{\delta}_t(\mathbf{x}) > 0\right] \mid H_t\right] &= \mathbb{P}(A \mid H_t)\\
&\overset{(i)}{\leq} \mathbb{P}(B \mid H_t)\\
&= \mathbb{P}\left(\bigcup_{\mathbf{c}\in\mathcal{C}}\{\{f(\mathbf{x},\mathbf{c}) < \ell_t(\mathbf{x},\mathbf{c})\}\cup\{f(\mathbf{x},\mathbf{c}) > u_t(\mathbf{x},\mathbf{c})\}\} \mid H_t\right)\\
&\overset{(ii)}{\leq} \sum_{\mathbf{c}\in\mathcal{C}}\mathbb{P}\left(\{f(\mathbf{x},\mathbf{c}) < \ell_t(\mathbf{x},\mathbf{c})\} \mid H_t\right)\\
&\quad + \mathbb{P}\left(\{f(\mathbf{x},\mathbf{c}) > u_t(\mathbf{x},\mathbf{c})\} \mid H_t\right)\\
&\overset{(iii)}{\leq} 2|\mathcal{C}|\mathbb{P}\left(X < 0\right)
\end{aligned}
\tag{25}
$$

where $(i)$ follows since, by the definition of $\hat{\delta}_t(\mathbf{x})$, if $\mathbf{f}(\mathbf{x})$ is inside the confidence bounds, $\delta_t(\mathbf{x}) - \hat{\delta}_t(\mathbf{x}) \leq 0$, therefore $B^c \subseteq A^c$ and $A \subseteq B$; $(ii)$ applies the union bound; and $(iii)$ follows by defining the random variable $X$ distributed as $\mathcal{N}(\sqrt{\beta_t}, 1)$ and recognizing that, conditioned on $H_t$, both $f(\mathbf{x},\mathbf{c}) - \ell_t(\mathbf{x},\mathbf{c})$ and $u_t(\mathbf{x},\mathbf{c}) - f(\mathbf{x},\mathbf{c})$ are distributed as $\mathcal{N}(\sqrt{\beta_t}\sigma_{t-1}(\mathbf{x},\mathbf{c}), \sigma_{t-1}^2(\mathbf{x},\mathbf{c}))$. Using $\Phi$ to denote the distribution function of the standard normal random variable,

$$
\begin{aligned}
\mathbb{P}\left(X < 0\right) &= \mathbb{P}\left(X - \sqrt{\beta_t} < -\sqrt{\beta_t}\right)\\
&= \Phi\left(-\sqrt{\beta_t}\right)\\
&\overset{(i)}{\leq} \frac{1}{2}e^{-\frac{\beta_t}{2}}\\
&= \frac{\sqrt{2\pi}}{(t^2 + 1)|\tilde{\mathcal{X}}_t||\mathcal{C}|}
\end{aligned}
$$

where $(i)$ follows from the Chernoff bound (Mastin & Jaillet, 2013). Combining this result with Equation 25 completes the proof. $\square$

## A.6 Proof of Lemma A.5

**Lemma A.5.** *Under the assumptions and definitions of Lemma A.3, for any* $\mathbf{x} \in \mathcal{X}$ *and* $t \leq T$,

$$\mathbb{E}_{H_t}\left[-\hat{\delta}_t(\mathbf{x})\right] \leq 2\left(\sqrt{\beta_t} + \sqrt{2\ln|\mathcal{C}|}\right) .$$

*Proof.* Define

$$
\begin{aligned}
\hat{\mathbf{q}}^h &:= \operatorname*{arg\,min}_{\mathbf{q}\in\mathcal{U}(\mathbf{p}_t, d, \epsilon_t + h)} \mathbf{q}^\top \boldsymbol{\varphi}_t(\mathbf{x})\\
\hat{\mathbf{q}} &:= \operatorname*{arg\,min}_{\mathbf{q}\in\mathcal{U}(\mathbf{p}_t, d, \epsilon_t)} \mathbf{q}^\top \boldsymbol{\varphi}_t(\mathbf{x}) .
\end{aligned}
$$

$$
\begin{aligned}
-\hat{\delta}_t(\mathbf{x}) &= \lim_{h\to 0}\frac{1}{h}\left((\hat{\mathbf{q}} - \hat{\mathbf{q}}^h)^\top \boldsymbol{\varphi}_t(\mathbf{x})\right)\\
&\overset{(i)}{=} \varphi_t(\mathbf{x})_{i^+} - \varphi_t(\mathbf{x})_{i^-}
\end{aligned}
$$

$$\leq \max_{\mathbf{c}\in\mathcal{C}} \varphi_t(\mathbf{x}, \mathbf{c}) - \min_{\mathbf{c}\in\mathcal{C}} \varphi_t(\mathbf{x}, \mathbf{c})$$

$$\leq \max_{\mathbf{c}\in\mathcal{C}} u_t(\mathbf{x}, \mathbf{c}) - \min_{\mathbf{c}\in\mathcal{C}} \ell_t(\mathbf{x}, \mathbf{c})$$

$$= \max_{\mathbf{c}\in\mathcal{C}} \left( \mu_{t-1}(\mathbf{x}, \mathbf{c}) + \sqrt{\beta_t}\sigma_{t-1}(\mathbf{x}, \mathbf{c}) \right) - \min_{\mathbf{c}\in\mathcal{C}} \left( \mu_{t-1}(\mathbf{x}, \mathbf{c}) - \sqrt{\beta_t}\sigma_{t-1}(\mathbf{x}, \mathbf{c}) \right)$$

$$\overset{(ii)}{\leq} 2\sqrt{\beta_t} + \max_{\mathbf{c}\in\mathcal{C}} \mu_{t-1}(\mathbf{x}, \mathbf{c}) - \min_{\mathbf{c}\in\mathcal{C}} \mu_{t-1}(\mathbf{x}, \mathbf{c})$$

where $(i)$ follows from Lemma A.12; and $(ii)$ follows since $k(\mathbf{x}, \mathbf{c}; \mathbf{x}, \mathbf{c}) = 1$ and hence the posterior variance $\sigma_{t-1}(\mathbf{x}, \mathbf{c}) \leq 1$ for all $t$, $\mathbf{x}$ and $\mathbf{c}$. Taking expectations,

$$\mathbb{E}_{H_t}\left[ -\hat{\delta}_t(\mathbf{x}) \right] \leq 2\sqrt{\beta_t} + \mathbb{E}_{H_t}\left[ \max_{\mathbf{c}\in\mathcal{C}} \mu_{t-1}(\mathbf{x}, \mathbf{c}) \right] - \mathbb{E}_{H_t}\left[ \min_{\mathbf{c}\in\mathcal{C}} \mu_{t-1}(\mathbf{x}, \mathbf{c}) \right] \tag{26}$$

We now bound $\mathbb{E}_{H_t}\left[ \max_{\mathbf{c}\in\mathcal{C}} \mu_{t-1}(\mathbf{x}, \mathbf{c}) \right]$. Recall the closed form of the posterior mean $\mu_{t-1}(\mathbf{x}, \mathbf{c})$:

$$\mu_{t-1}(\mathbf{x}, \mathbf{c}) = \mathbf{k}_{t-1}(\mathbf{x}, \mathbf{c})^\top (\mathbf{K}_{t-1} + \sigma^2 \mathbf{I})^{-1} \mathbf{y}_{t-1}$$

$$= \mathbf{a}_{t-1}(\mathbf{x}, \mathbf{c})^\top \mathbf{y}_{t-1}$$

where we have defined $\mathbf{a}_{t-1}(\mathbf{x}, \mathbf{c}) := (\mathbf{K}_{t-1} + \sigma^2 \mathbf{I})^{-1} \mathbf{k}_{t-1}(\mathbf{x}, \mathbf{c})$. Now, denote the relevant random variables in $H_t$ as $\mathbf{X}_{t-1} = \{\mathbf{x}_\tau\}_{\tau=1}^{t-1}$ and $\mathbf{C}_{t-1} = \{\mathbf{c}_\tau\}_{\tau=1}^{t-1}$, with $\mathbf{y}_{t-1}$ having its usual definition.

$$\mathbb{E}_{H_t}\left[ \max_{\mathbf{c}\in\mathcal{C}} \mu_{t-1}(\mathbf{x}, \mathbf{c}) \right] = \mathbb{E}_{\mathbf{X}_{t-1}, \mathbf{C}_{t-1}}\left[ \mathbb{E}_{\mathbf{y}_{t-1}}\left[ \max_{\mathbf{c}\in\mathcal{C}} \mathbf{a}_{t-1}(\mathbf{x}, \mathbf{c})^\top \mathbf{y}_{t-1} \mid \mathbf{X}_{t-1}, \mathbf{C}_{t-1} \right] \right] . \tag{27}$$

Since $\mathbf{y}_{t-1}$ conditioned on $\mathbf{X}_{t-1}, \mathbf{C}_{t-1}$ is distributed as $\mathcal{N}(\mathbf{0}, \mathbf{K}_{t-1} + \sigma^2 \mathbf{I})$, for each $\mathbf{c} \in \mathcal{C}$, $\mathbf{a}_{t-1}(\mathbf{x}, \mathbf{c})^\top \mathbf{y}_{t-1}$ is distributed as $\mathcal{N}(0, \mathbf{a}_{t-1}(\mathbf{x}, \mathbf{c})^\top (\mathbf{K}_{t-1} + \sigma^2 \mathbf{I})\mathbf{a}_{t-1}(\mathbf{x}, \mathbf{c})) = \mathbf{k}_{t-1}(\mathbf{x}, \mathbf{c})^\top (\mathbf{K}_{t-1} + \sigma^2 \mathbf{I})^{-1} \mathbf{k}_{t-1}(\mathbf{x}, \mathbf{c}) = 1 - \sigma_{t-1}^2(\mathbf{x}, \mathbf{c})$. Hence,

$$\mathbb{E}_{\mathbf{y}_{t-1}}\left[ \max_{\mathbf{c}\in\mathcal{C}} \mathbf{a}_{t-1}(\mathbf{x}, \mathbf{c})^\top \mathbf{y}_{t-1} \mid \mathbf{X}_{t-1}, \mathbf{C}_{t-1} \right] = \mathbb{E}\left[ Z \right] \tag{28}$$

$$Z := \max_{\mathbf{c}\in\mathcal{C}} A_{\mathbf{c}}$$

$$A_{\mathbf{c}} \sim \mathcal{N}(0, 1 - \sigma_{t-1}^2(\mathbf{x}, \mathbf{c})) .$$

The random variables $A_{\mathbf{c}}$ are in general not independent. However, the following method[3] to upper bound $\mathbb{E}\left[ Z \right]$ does not assume independence. Letting $b$ be some strictly positive constant,

$$e^{b\mathbb{E}[Z]} \overset{(i)}{\leq} \mathbb{E}\left[ e^{bZ} \right]$$

$$= \mathbb{E}\left[ \max_{\mathbf{c}\in\mathcal{C}} e^{bA_{\mathbf{c}}} \right]$$

$$\leq \sum_{\mathbf{c}\in\mathcal{C}} \mathbb{E}\left[ e^{bA_{\mathbf{c}}} \right]$$

$$\overset{(ii)}{=} |\mathcal{C}| e^{\frac{1}{2}b^2(1 - \sigma_{t-1}^2(\mathbf{x}, \mathbf{c}))^2}$$

$$\overset{(iii)}{\leq} |\mathcal{C}| e^{\frac{1}{2}b^2}$$

where $(i)$ applies Jensen's inequality; $(ii)$ uses the definition of the moment generating function of a Gaussian random variable; and $(iii)$ follows since the posterior variance $0 \leq \sigma_{t-1}^2(\mathbf{x}, \mathbf{c}) \leq 1$. Setting $b = \sqrt{2 \ln |\mathcal{C}|}$ and rearranging the terms, we arrive at

$$\mathbb{E}\left[ Z \right] \leq \sqrt{2 \ln |\mathcal{C}|} .$$

Combine this result with Equation 27 and Equation 28 to conclude that

$$\mathbb{E}_{H_t}\left[ \max_{\mathbf{c}\in\mathcal{C}} \mu_{t-1}(\mathbf{x}, \mathbf{c}) \right] \leq \sqrt{2 \ln |\mathcal{C}|} .$$

---

[3] due to Sivaraman at `https://math.stackexchange.com/q/89147`.

Finally, observe that $\min_{\mathbf{c} \in \mathcal{C}} \mu_{t-1}(\mathbf{x}, \mathbf{c}) = \max_{\mathbf{c} \in \mathcal{C}} -\mu_{t-1}(\mathbf{x}, \mathbf{c})$. Since $\mathbf{y}_{t-1}$ conditioned on $\mathbf{X}_{t-1}, \mathbf{C}_{t-1}$ has mean $\mathbf{0}$, $-\mu_{t-1}(\mathbf{x}, \mathbf{c})$ has the same distribution as $\mu_{t-1}(\mathbf{x}, \mathbf{c})$. We use this fact to conclude from Equation 26 with our previous result that

$$\mathbb{E}_{H_t} \left[ -\hat{\delta}_t(\mathbf{x}) \right] \leq 2 \left( \sqrt{\beta_t} + \sqrt{2 \ln |\mathcal{C}|} \right)$$

which completes the proof. $\qquad \square$

## A.7 PROOF OF LEMMA A.6

**Lemma A.6.** *If the kernel $k$ is the squared exponential kernel or a Matérn kernel with $\nu > 2$, then for all $t \leq T$ and $\mathbf{x} \in \mathcal{X}$,*

$$\mathbb{E}\left[|v_t(\mathbf{x}) - v_t([\mathbf{x}]_t)|\right] \leq 2\mathbb{E}\left[B\epsilon_t\right] + \frac{1}{2t^2} \tag{29}$$

*where $B$ is a complexity parameter that depends on distribution distance $d$ and $f$. For example, $B$ has the following specific forms for maximum mean discrepancy (MMD) and total variation (TV):*

$$B := \begin{cases} \max_{\mathbf{x} \in \mathcal{X}} \|\mathbf{f}(\mathbf{x})\|_{M^{-1}} & \text{if } d = \text{MMD}, \\ \max_{\mathbf{x} \in \mathcal{X}} \|\mathbf{f}(\mathbf{x})\| & \text{if } d = \text{TV}. \end{cases}$$

*Proof.* For this proof, define

$$\mathbf{q}_1 := \underset{\mathbf{q} \in \mathcal{U}(\mathbf{p}_t, d, \epsilon_t)}{\arg\min} \mathbf{q}^\top \mathbf{f}(\mathbf{x})$$

$$\mathbf{q}_2 := \underset{\mathbf{q} \in \mathcal{U}(\mathbf{p}_t, d, \epsilon_t)}{\arg\min} \mathbf{q}^\top \mathbf{f}([\mathbf{x}]_t)$$

$$\begin{aligned}
\mathbb{E}\left[|v_t(\mathbf{x}) - v_t([\mathbf{x}]_t)|\right] &= \mathbb{E}\left[\left|\mathbf{q}_1^\top \mathbf{f}(\mathbf{x}) - \mathbf{q}_2^\top \mathbf{f}([\mathbf{x}]_t)\right|\right] \\
&= \mathbb{E}\left[\left|(\mathbf{q}_1 - \mathbf{q}_2)^\top \mathbf{f}(\mathbf{x}) + \mathbf{q}_2^\top (\mathbf{f}(\mathbf{x}) - \mathbf{f}([\mathbf{x}]_t))\right|\right] \\
&\leq \mathbb{E}\left[\left|(\mathbf{q}_1 - \mathbf{q}_2)^\top \mathbf{f}(\mathbf{x})\right| + \left|\mathbf{q}_2^\top (\mathbf{f}(\mathbf{x}) - \mathbf{f}([\mathbf{x}]_t))\right|\right] \\
&\overset{(i)}{\leq} 2\mathbb{E}\left[B\epsilon_t\right] + \mathbb{E}\left[\left|\mathbf{q}_2^\top (\mathbf{f}(\mathbf{x}) - \mathbf{f}([\mathbf{x}]_t))\right|\right] \\
&\overset{(ii)}{\leq} 2\mathbb{E}\left[B\epsilon_t\right] + \frac{1}{2t^2}
\end{aligned}$$

where $(i)$ follows the proof of Theorem 4 in Tay et al. (2022) (Equation 81 onwards) since $\mathbf{q}_1$ and $\mathbf{q}_2$ are at most $2\epsilon_t$ distance from each other via the triangular inequality with $\mathbf{p}_t$; and $(ii)$ follows from Lemma A.8, which completes the proof. $\qquad \square$

## A.8 PROOF OF LEMMA A.7

**Lemma A.7.** *If the following assumptions hold:*

1. *The kernel $k$ is the squared exponential kernel or a Matérn kernel with $\nu > 2$;*

2. *the distribution distance $d$ is the total variation (TV) distance,*

*then for all $t \leq T$ and $\mathbf{x} \in \mathcal{X}$,*

$$\mathbb{E}\left[|\delta_t(\mathbf{x}) - \delta_t([\mathbf{x}]_t)|\right] \leq \frac{1}{t^2}. \tag{30}$$

*Proof.* For this proof, define

$$\mathbf{q}_1^h := \underset{\mathbf{q} \in \mathcal{U}(\mathbf{p}_t, d, \epsilon_t + h)}{\arg\min} \mathbf{q}^\top \mathbf{f}(\mathbf{x})$$

$$\mathbf{q}_1 := \underset{\mathbf{q} \in \mathcal{U}(\mathbf{p}_t, d, \epsilon_t)}{\arg\min} \mathbf{q}^\top \mathbf{f}(\mathbf{x})$$

$$\mathbf{q}_2^h := \arg\min_{\mathbf{q}\in\mathcal{U}(\mathbf{p}_t,d,\epsilon_t+h)} \mathbf{q}^\top \mathbf{f}([\mathbf{x}]_t)$$

$$\mathbf{q}_2 := \arg\min_{\mathbf{q}\in\mathcal{U}(\mathbf{p}_t,d,\epsilon_t)} \mathbf{q}^\top \mathbf{f}([\mathbf{x}]_t)$$

$$
\begin{aligned}
\mathbb{E}\left[|\delta_t(\mathbf{x}) - \delta_t([\mathbf{x}]_t)|\right] &= \mathbb{E}\left[\left|\lim_{h\to 0}\frac{1}{h}\left((\mathbf{q}_1^h - \mathbf{q}_1)^\top \mathbf{f}(\mathbf{x}) - (\mathbf{q}_2^h - \mathbf{q}_2)^\top \mathbf{f}([\mathbf{x}]_t)\right)\right|\right] \\
&\overset{(i)}{=} \mathbb{E}\left[\left|\lim_{h\to 0}\frac{1}{h}\left(h(\mathbf{e}_{i^-} - \mathbf{e}_{i^+})^\top \mathbf{f}(\mathbf{x}) - h(\mathbf{e}_{j^-} - \mathbf{e}_{j^+})^\top \mathbf{f}([\mathbf{x}]_t)\right)\right|\right] \\
&= \mathbb{E}\left[\left|(\mathbf{e}_{i^-} - \mathbf{e}_{i^+})^\top \mathbf{f}(\mathbf{x}) - (\mathbf{e}_{j^-} - \mathbf{e}_{j^+})^\top \mathbf{f}([\mathbf{x}]_t)\right|\right] \\
&= \mathbb{E}\left[\left|f(\mathbf{x})_{i^-} - f(\mathbf{x})_{i^+} - f([\mathbf{x}]_t)_{j^-} + f([\mathbf{x}]_t)_{j^+}\right|\right] \\
&\leq \mathbb{E}\left[\left|f(\mathbf{x})_{j^-} - f([\mathbf{x}]_t)_{j^-} + f([\mathbf{x}]_t)_{j^+} - f(\mathbf{x})_{i^+}\right|\right] \\
&\overset{(ii)}{\leq} \mathbb{E}\left[\left|(f(\mathbf{x})_{j^-} - f([\mathbf{x}]_t)_{j^-}) + (f([\mathbf{x}]_t)_o - f(\mathbf{x})_o)\right|\right] \\
&\overset{(iii)}{\leq} \frac{1}{t^2}
\end{aligned}
$$

where $(i)$ follows from applying Lemma A.12 and defining $i^- := \arg\min_i f(\mathbf{x})_i$, $i^+ := \arg\max_i\{f(\mathbf{x})_i : q_{1,i} > 0\}$, and similarly defining $j^-$ and $j^+$ with respect to $\mathbf{f}([\mathbf{x}]_t)$ and $\mathbf{q}_2$; and $(ii)$ follows from applying Lemma A.13 where $o$ is some index in $[n]$; and $(iii)$ follows from applying Lemma A.8, which completes the proof. $\qquad\square$

## A.9 PROOF OF LEMMA A.8

**Lemma A.8.** *If the kernel $k$ is the squared exponential kernel or a Matérn kernel with $\nu > 2$, then for all $t \leq T$, $\mathbf{x} \in \mathcal{X}$ and $\mathbf{c} \in \mathcal{C}$,*

$$\mathbb{E}\left[|f(\mathbf{x},\mathbf{c}) - f([\mathbf{x}]_t,\mathbf{c})|\right] \leq \frac{1}{2t^2}. \tag{31}$$

*Proof.* Recall that, under the assumption that the kernel $k$ is the squared exponential kernel or a Matérn kernel with $\nu > 2$, GP sample paths $f$ satisfy the high probability bound $\mathbb{P}(\sup_{(\mathbf{x},\mathbf{c})\in\mathcal{X}\times\mathcal{C}}|\frac{\partial f(\mathbf{x},\mathbf{c})}{\partial x_i}| > J) \leq ae^{-(J/b)^2}$ for all $i \in \{1,...,m\}$ for some kernel-dependent constants $a$ and $b$ (Ghosal & Roy, 2006). The lemma then follows directly from the proof of Lemma 12 in Kandasamy et al. (2018), by choosing $L$ in that proof to be $\sup_{i=1,...,m}\sup_{(\mathbf{x},\mathbf{c})\in\mathcal{X}\times\mathcal{C}}|\frac{\partial f(\mathbf{x},\mathbf{c})}{\partial x_i}|$, and noting that the presence of the context variable does not affect the $\ell^1$ norm since it is the same in $(\mathbf{x},\mathbf{c})$ and $([\mathbf{x}]_t,\mathbf{c})$. $\qquad\square$

## A.10 PROOF OF LEMMA A.9

**Lemma A.9.** *If the kernel $k$ is the squared exponential kernel or a Matérn kernel with $\nu > 2$, then for all $t \leq T$, $\mathbf{x} \in \mathcal{X}$ and $\mathbf{c} \in \mathcal{C}$,*

$$\left|\sigma_{t-1}^2(\mathbf{x},\mathbf{c}) - \sigma_{t-1}^2([\mathbf{x}]_t,\mathbf{c})\right| \leq \frac{4Lm}{\tau_t}$$

*where $L$ is a kernel-dependent constant.*

*Proof.* The proof relies on the following lemma developed in Shekhar & Javidi (2020) and re-stated as in Li & Scarlett (2022):

**Lemma A.10** (Shekhar & Javidi (2020), Proposition 1 and Remark 5). *For any $f$ in the RKHS $\mathcal{H}_k$ associated with kernel $k$ with RKHS norm $\|f\|_{\mathcal{H}_k}$ less than $\Psi$, if the kernel $k$ is the squared exponential kernel or a Matérn kernel with $\nu > 2$, and $\Psi$ is constant, then $f$ is guaranteed to be Lipschitz continuous with some constant $L$ depending only on the kernel parameters.*

We will first show that the terms making up the GP posterior covariance at any iteration $t$ are in $\mathcal{H}_k$ with RKHS norms bounded by fixed constants, as in the proof of Lemma F.2 in Vakili et al. (2022). With the above lemma, this implies their Lipschitz continuity, which allows the discretization error to be bounded.

To simplify notation, define the joint decision-context space $\mathcal{Z} := \mathcal{X} \times \mathcal{C}$, and joint decision-context vectors $\mathbf{z} \in \mathcal{Z}$, $\mathbf{z} = (\mathbf{x}, \mathbf{c})$ for some $\mathbf{x}$ and $\mathbf{c}$. The kernel $k$ originally denoted $k(\mathbf{x}, \mathbf{c}; \mathbf{x}', \mathbf{c}')$ can now be simply denoted $k(\mathbf{z}, \mathbf{z}')$, and the kernel vectors and matrices as defined at the start of Sec. 4 can be cast in terms of $\mathbf{z}$ similarly. The posterior variance at iteration $t$ $\sigma_{t-1}^2(\mathbf{x}, \mathbf{c})$ Equation 8 can now be simply denoted $\sigma_{t-1}^2(\mathbf{z})$. Let $[\mathbf{z}]_t$ denote $([\mathbf{x}]_t, \mathbf{c})$.

Define

$$q(\mathbf{z}, \cdot) := \mathbf{v}(\mathbf{z})^\top \mathbf{k}_{t-1}(\cdot)$$
$$\mathbf{v}(\cdot) := (\mathbf{K}_{t-1} + \sigma^2 \mathbf{I})^{-1} \mathbf{k}_{t-1}(\cdot)$$

Note that $\sigma_{t-1}^2(\mathbf{z}) = k(\mathbf{z}, \mathbf{z}) - q(\mathbf{z}, \mathbf{z})$.

$$\|k(\mathbf{z}, \cdot)\|_{\mathcal{H}_k} = \sqrt{\langle k(\mathbf{z}, \cdot), k(\mathbf{z}, \cdot)\rangle_{\mathcal{H}_k}} = \sqrt{k(\mathbf{z}, \mathbf{z})} = 1$$

by the reproducing property and the assumption that $k(\mathbf{z}, \mathbf{z}) = 1$ for all $\mathbf{z}$.

$$\begin{aligned}
\|q(\mathbf{z}, \cdot)\|_{\mathcal{H}_k}^2 &= \left\langle \mathbf{v}(\mathbf{z})^\top \mathbf{k}_{t-1}(\cdot), \mathbf{v}(\mathbf{z})^\top \mathbf{k}_{t-1}(\cdot) \right\rangle_{\mathcal{H}_k} \\
&\overset{(i)}{=} \mathbf{v}(\mathbf{z})^\top \mathbf{K}_{t-1} \mathbf{v}(\mathbf{z}) \\
&= \mathbf{v}(\mathbf{z})^\top \left( \mathbf{K}_{t-1} + \sigma^2 \mathbf{I} - \sigma^2 \mathbf{I} \right) \mathbf{v}(\mathbf{z}) \\
&= \mathbf{v}(\mathbf{z})^\top \left( \mathbf{K}_{t-1} + \sigma^2 \mathbf{I} \right) \mathbf{v}(\mathbf{z}) - \mathbf{v}(\mathbf{z})^\top \left( \sigma^2 \mathbf{I} \right) \mathbf{v}(\mathbf{z}) \\
&= q(\mathbf{z}, \mathbf{z}) - \sigma^2 \mathbf{v}(\mathbf{z})^\top \mathbf{v}(\mathbf{z}) \\
&\leq q(\mathbf{z}, \mathbf{z}) \\
&\overset{(ii)}{\leq} k(\mathbf{z}, \mathbf{z}) = 1
\end{aligned}$$

where $(i)$ follows from the reproducing property, and $(ii)$ follows since the GP posterior variance $\sigma_{t-1}^2(\mathbf{z})$ could be negative if this inequality were not true for all $\mathbf{z}$.

Since $\|k(\mathbf{z}, \cdot)\|_{\mathcal{H}_k} \leq 1$ and $\|q(\mathbf{z}, \cdot)\|_{\mathcal{H}_k} \leq 1$, by Lemma A.10, $k(\mathbf{z}, \cdot)$ and $q(\mathbf{z}, \cdot)$ are Lipschitz continuous with some kernel-dependent constant $L$. Since $\mathbf{z}$ was arbitrarily chosen, $k([\mathbf{z}]_t, \cdot)$ and $q([\mathbf{z}]_t, \cdot)$ are also $L$-Lipschitz continuous. Recall that the choice of discretization $\tilde{\mathcal{X}}_t$ leads to $\|\mathbf{z} - [\mathbf{z}]_t\|_2 \leq \frac{m}{\tau_t}$. Using these facts,

$$\begin{aligned}
|k(\mathbf{z}, \mathbf{z}) - k([\mathbf{z}]_t, [\mathbf{z}]_t)| &\leq |k(\mathbf{z}, \mathbf{z}) - k([\mathbf{z}]_t, \mathbf{z})| + \frac{Lm}{\tau_t} \\
&\leq |k(\mathbf{z}, \mathbf{z}) - k(\mathbf{z}, \mathbf{z})| + \frac{2Lm}{\tau_t} \\
&= \frac{2Lm}{\tau_t} \ .
\end{aligned}$$

By the same argument, $|q(\mathbf{z}, \mathbf{z}) - q([\mathbf{z}]_t, [\mathbf{z}]_t)| \leq \frac{2Lm}{\tau_t}$. We thus conclude with

$$\begin{aligned}
\left| \sigma_{t-1}^2(\mathbf{z}) - \sigma_{t-1}^2([\mathbf{z}]_t) \right| &= |k(\mathbf{z}, \mathbf{z}) - q(\mathbf{z}, \mathbf{z}) - k([\mathbf{z}]_t, [\mathbf{z}]_t) + q([\mathbf{z}]_t, [\mathbf{z}]_t)| \\
&\leq |k(\mathbf{z}, \mathbf{z}) - k([\mathbf{z}]_t, [\mathbf{z}]_t)| + |q([\mathbf{z}]_t, [\mathbf{z}]_t) - q(\mathbf{z}, \mathbf{z})| \\
&\leq \frac{4Lm}{\tau_t}
\end{aligned}$$

which concludes the proof. $\qquad \square$

### A.11 PROOF OF LEMMA A.11

**Lemma A.11.** *Without loss of generality, suppose the entries of $\mathbf{f} \in \mathbb{R}^n$ are ordered such that $f_1 \geq f_2 \geq ... \geq f_n$. For a given reference probability distribution $\mathbf{p}$ and margin $\epsilon$, let $k$ be the first*

*index such that $\sum_{i=1}^{k} p_i > \frac{\epsilon}{2}$. If $d$ is the total variation distance and $\mathbf{q}$ is a probability vector such that, if $k < n$,*

$$q_i = \begin{cases} 0, & \text{for } 1 \le i < k, \\ \sum_{j=1}^{k} p_j - \frac{\epsilon}{2}, & \text{for } i = k, \\ p_i, & \text{for } k+1 \le i < n, \\ p_n + \frac{\epsilon}{2}, & \text{for } i = n, \end{cases}$$

*or if $k = n$ or $k$ does not exist (i.e., $\sum_{i=1}^{n} p_i \le \frac{\epsilon}{2}$),*

$$q_i = \begin{cases} 0, & \text{for } 1 \le i < n, \\ 1, & \text{for } i = n, \end{cases}$$

*then $\mathbf{q} \in \arg\min_{\mathbf{q} \in \mathcal{U}(\mathbf{p}, d, \epsilon)} \mathbf{q}^\top \mathbf{f}$.*

*Proof.* $\mathbf{q}$ is the solution to the convex optimization problem

$$\min_{\mathbf{q} \in \mathbb{R}^n} \quad g_0(\mathbf{q}) := \mathbf{q}^\top \mathbf{f}$$

$$\text{s.t.} \quad g_i(\mathbf{q}) := -q_i \le 0, \quad 1 \le i \le n$$

$$g_{n+1}(\mathbf{q}) := \|\mathbf{q} - \mathbf{p}\|_1 - \epsilon \le 0$$

$$h(\mathbf{q}) := \mathbf{1}^\top \mathbf{q} = 1 .$$

$\mathbf{q}$ is optimal if there exists Lagrange multipliers $\boldsymbol{\lambda} \in \mathbb{R}^{n+1}, \boldsymbol{\lambda} \succeq \mathbf{0}$ and $\nu \in \mathbb{R}$ such that the Karush-Kuhn-Tucker (KKT) conditions are satisfied (Rockafellar, 1997, Theorem. 28.3). These conditions require the primal feasibility of $\mathbf{q}$ and the dual feasibility of $\boldsymbol{\lambda}$ as well as the following conditions on complementary slackness and stationarity:

$$\lambda_i g_i(\mathbf{q}) = 0, \quad 1 \le i \le n+1$$

$$\mathbf{0} \in \partial g_0(\mathbf{q}) + \sum_{i=1}^{n+1} \lambda_i \partial g_i(\mathbf{q}) + \nu \partial h(\mathbf{q})$$

where $\partial$ denotes the subdifferential.

We first verify that $\mathbf{q}$ is feasible. It is straightforward to see that $\mathbf{1}^\top \mathbf{q} = 1$ and $\mathbf{q} \succeq \mathbf{0}$ in all cases. We now show that $\|\mathbf{p} - \mathbf{q}\|_1 \le \epsilon$. In the case that $k < n$,

$$\|\mathbf{p} - \mathbf{q}\|_1 = \sum_{i=1}^{n} |p_i - q_i|$$

$$= \sum_{i=1}^{k-1} p_i + |p_k - q_k| + |p_n - q_n|$$

$$= \sum_{i=1}^{k-1} p_i + \left| p_k - \left( \sum_{i=1}^{k} p_i - \frac{\epsilon}{2} \right) \right| + \left| p_n - \left( p_n + \frac{\epsilon}{2} \right) \right|$$

$$= \sum_{i=1}^{k-1} p_i + \left| \frac{\epsilon}{2} - \sum_{i=1}^{k-1} p_i \right| + \frac{\epsilon}{2}$$

$$= \epsilon$$

where the last equality follows since $\sum_{i=1}^{k-1} p_i \le \frac{\epsilon}{2}$ by definition of $k$. In the case that $k = n$ or $k$ does not exist,

$$\|\mathbf{p} - \mathbf{q}\|_1 = \sum_{i=1}^{n-1} p_i + (1 - p_n)$$

$$= 2 \sum_{i=1}^{n-1} p_i$$

$$\leq \epsilon$$

where the last inequality follows since $\sum_{i=1}^{n-1} p_i \leq \frac{\epsilon}{2}$ since $k = n$ or $k$ does not exist.

Clearly, if $k = n$ or $k$ does not exist, $\mathbf{q}$ is optimal since $\mathbf{q}^\top \mathbf{f} = \min_i f_i$, which is the minimum expected value that any probability distribution can attain. We continue to verify the complementary slackness and stationarity conditions in the case that $k < n$. The subdifferentials are

$$\partial g_0(\mathbf{q}) = \{\mathbf{f}\}$$
$$\partial g_i(\mathbf{q}) = \{-\mathbf{e}_i\}, \quad 1 \leq i \leq n$$
$$\partial h(\mathbf{q}) = \{\mathbf{1}\}$$

where $\mathbf{e}_i$ is a vector with all entries equal to 0 except the $i$-th entry. The subdifferential of $g_{n+1}$ depends on $\mathbf{q} - \mathbf{p}$ which has the form

$$\mathbf{q} - \mathbf{p} = \begin{cases} -p_i, & \text{for } 1 \leq i < k, \\ \sum_{j=1}^{k-1} p_j - \frac{\epsilon}{2}, & \text{for } i = k, \\ 0 & \text{for } k+1 \leq i < n, \\ \frac{\epsilon}{2}, & \text{for } j = n. \end{cases}$$

The entries of the subdifferential of $g_{n+1}$ are then given by

$$\partial g_{n+1}(\mathbf{q})_i \in \begin{cases} \{-1\}, & \text{for } 1 \leq i < k, \\ \{-1\} \text{ if } \sum_{j=1}^{k-1} < \frac{\epsilon}{2}, [-1,1] \text{ otherwise} & \text{for } i = k, \\ [-1,1], & \text{for } k+1 \leq i < n, \\ \{1\}, & \text{for } i = n \end{cases}$$

Without loss of generality, we set the $k$-th entry of $\partial g_{n+1}(\mathbf{q})$ to be $-1$.

To satisfy complementary slackness, since $g_i < 0$ for $k \leq i \leq n$,

$$\lambda_i = 0, \quad \text{for } k \leq i \leq n.$$

It remains to choose the values of $\lambda_i$ for $1 \leq i < k$, $\lambda_{n+1}$, and $\nu$ such that they satisfy dual feasibility and stationarity.

The stationarity condition can be seen as a system of $n + 1$ equations. In addition to the dual variables, we introduce auxiliary variables $a_i \in [-1, 1]$ for $k + 1 \leq i < n$, indicating the specific value of the subgradient at index $i$ that satisfy the stationarity equations since the subdifferential for these indices is the range $[-1, 1]$. The stationarity equations then become

$$f_i - \lambda_i - \lambda_{n+1} + \nu = 0, \quad \text{for } 1 \leq i < k, \tag{32}$$
$$f_k - \lambda_{n+1} + \nu = 0, \tag{33}$$
$$f_i + \lambda_{n+1} a_i + \nu = 0, \quad \text{for } k+1 \leq i < n, \tag{34}$$
$$f_n + \lambda_{n+1} + \nu = 0. \tag{35}$$

To satisfy Equation 33 and Equation 35, choose

$$\nu = -f_n - \lambda_{n+1}$$
$$\lambda_{n+1} = \frac{f_k - f_n}{2} \geq 0.$$

For $1 \leq i < k$, to satisfy Equation 32, choose each $\lambda_i$ to be

$$\lambda_i = f_i - \lambda_{n+1} + \nu$$
$$= f_i - f_k \geq 0.$$

For $k + 1 \leq i < n$, to satisfy Equation 34, choose each $a_i$ to be

$$a_i = \frac{-\nu - f_i}{\lambda_{n+1}}$$
$$= \frac{f_k - f_i}{f_k - f_n} - \frac{f_i - f_n}{f_k - f_n} \in [-1, 0]$$

where the last inclusion arises since $\frac{f_k - f_i}{f_k - f_n} \in [0, 1]$ and $\frac{f_i - f_n}{f_k - f_n} \in [0, 1]$. Since all KKT conditions are satisfied, $\mathbf{q}$ is optimal and the proof is complete. $\qquad\square$

## A.12 PROOF OF LEMMA A.12

**Lemma A.12.** *If $d$ is the total variation distance and $\mathbf{q}_\epsilon := \arg\min_{\mathbf{q} \in \mathcal{U}_d(\mathbf{p}, \epsilon)} \mathbf{q}^\top \mathbf{f}$, then for all $h \leq q_{\epsilon,k}$ where $k := \arg\max_j \{f_j : q_{\epsilon,j} > 0\}$,*

$$\mathbf{q}_{\epsilon+h} - \mathbf{q}_\epsilon = h(\mathbf{e}_i - \mathbf{e}_k)$$

*where $\mathbf{e}_k$ is a vector with all entries equal to $0$ except the $k$-th entry and $i = \arg\min_i f_i$.*

*Proof.* The proof uses the closed form of $\mathbf{q}_\epsilon$ prescribed in Lemma A.11. Observe that the index $k$ defined here is the same as that defined in Lemma A.11 when it exists, and that when $h \leq q_{\epsilon,k}$, the index $k$ for $\mathbf{q}_{\epsilon+h}$ is the same as that for $\mathbf{q}_\epsilon$. The result follows from subtracting the closed form of $\mathbf{q}_{\epsilon+h}$ from $\mathbf{q}_\epsilon$. When $k$ does not exist in Lemma A.11, then $\mathbf{q}_{\epsilon+h} = \mathbf{q}_\epsilon$ and $k = i$ in this Lemma, thus the result still holds. $\square$

## A.13 PROOF OF LEMMA A.13

**Lemma A.13.** *Let $\boldsymbol{\varphi}$ and $\mathbf{f}$ be vectors in $\mathbb{R}^n$, and suppose without loss of generality that the entries of $\boldsymbol{\varphi}$ are ordered such that $\varphi_1 \geq \varphi_2 \geq ... \geq \varphi_n$. Let $\hat{\mathbf{q}} := \arg\min_{\mathbf{q} \in \mathcal{U}_d(\mathbf{p}, \epsilon)} \mathbf{q}^\top \boldsymbol{\varphi}$ and $\mathbf{q} := \arg\min_{\mathbf{q} \in \mathcal{U}_d(\mathbf{p}, \epsilon)} \mathbf{q}^\top \mathbf{f}$, with $d$ being the total variation distance. Let $i := \arg\max_i \{\varphi_j : \hat{q}_j > 0\}$ and $j := \arg\max_i \{f_i : q_i > 0\}$. Then, there exists some index $\ell \in [n]$ such that*

$$f_j - \varphi_i \leq f_\ell - \varphi_\ell .$$

*Proof.* The result is trivial when $i = j$. The rest of this proof focuses on the case of $i \neq j$. The proof uses the closed form of $\mathbf{q}_\epsilon$ prescribed in Lemma A.11. Observe that the index $i$ defined here is the same as $k$ defined in Lemma A.11 when it exists. When $k$ does not exist in Lemma A.11, then $j = \arg\min_i f_i$ and choosing $\ell = i$ satisfies the inequality. We now focus on the case when it exists.

Suppose $j > i$. Then, by the ordering of entries in $\boldsymbol{\varphi}$, choosing $\ell = j$ satisfies the inequality. If $j < i$, then we require the existence of an index $\ell$ such that $\ell \geq i$ and $f_\ell \geq f_j$. As a proof of contradiction, suppose such an index does not exist, i.e., for all $m \geq i$, $f_m < f_j$. Then, by Lemma A.11, $q_m \geq p_m$ for all $m \geq i$. Let $o := \arg\min_k f_k$. Again by Lemma A.11, $q_o = p_o + \frac{\epsilon}{2}$. With these facts, consider a lower bound on the amount of probability mass in $\mathbf{q}$, attained by supposing $o \geq i$ and $q_k = 0$ for all $k < i, k \neq j$:

$$\sum_{k=1}^n q_k \geq q_j + \sum_{k=i}^n q_k + \frac{\epsilon}{2}$$

$$= q_j + \sum_{k=i}^n p_k + \frac{\epsilon}{2}$$

$$> \sum_{k=i}^n p_k + \frac{\epsilon}{2} = 1$$

where the last equality uses Lemma A.11 on $\hat{\mathbf{q}}$. We have arrived at a contradiction, since the amount of probability mass in $\mathbf{q}$ must sum to 1. Thus, when $j < i$, there must always exist an index $\ell$ such that $\ell \geq i$ and $f_\ell \geq f_j$. Choosing this index $\ell$ satisfies the inequality and completes the proof. $\square$

## A.14 PROOF OF PROPOSITION 4.2

**Proposition 4.2.** *If the distribution distance $d$ is the total variation (TV) distance, then the derivative upper confidence bound $\hat{\delta}_t(\mathbf{x})$ defined in Equation 11 exists.*

*Proof.* Recall the definition of $\hat{\delta}_t(\mathbf{x})$:

$$\hat{\delta}_t(\mathbf{x}) := \max_{\boldsymbol{\ell}_t(\mathbf{x}) \preceq \boldsymbol{\varphi} \preceq \mathbf{u}_t(\mathbf{x})} \lim_{h \to 0} \frac{1}{h} \left( \min_{\mathbf{q} \in \mathcal{U}(\mathbf{p}_t, d, \epsilon_t + h)} \mathbf{q}^\top \boldsymbol{\varphi} - \min_{\mathbf{q} \in \mathcal{U}(\mathbf{p}_t, d, \epsilon_t)} \mathbf{q}^\top \boldsymbol{\varphi} \right) .$$

The extreme value theorem for metric spaces states that a continuous function attains its maximum on a compact set. Thus, a sufficient condition to guarantee the existence of $\hat{\delta}_t(\mathbf{x})$ is that $\lim_{h \to 0} \frac{1}{h} \left( \min_{\mathbf{q} \in \mathcal{U}(\mathbf{p}_t, d, \epsilon_t + h)} \mathbf{q}^\top \boldsymbol{\varphi} - \min_{\mathbf{q} \in \mathcal{U}(\mathbf{p}_t, d, \epsilon_t)} \mathbf{q}^\top \boldsymbol{\varphi} \right)$ is a continuous function.

From an application of Danskin's theorem (Bertsekas, 1999, Prop. B.25), we obtain that the optimal value of a distributionally robust convex optimization problem $\min_{\mathbf{q} \in \mathcal{U}(\mathbf{p}_t, d, \epsilon_t)} \mathbf{q}^\top \boldsymbol{\varphi}$ is a convex function of $\boldsymbol{\varphi} \in \mathbb{R}^n$. This implies that the function is Lipschitz continuous on the compact set $\mathcal{A} := \{\boldsymbol{\varphi} : \boldsymbol{\ell}_t(\mathbf{x}) \preceq \boldsymbol{\varphi} \preceq \mathbf{u}_t(\mathbf{x})\}$ (Rockafellar, 1997, Theorem. 10.4), a stronger condition of continuity. Thus, $\frac{1}{h} \left( \min_{\mathbf{q} \in \mathcal{U}(\mathbf{p}_t, d, \epsilon_t + h)} \mathbf{q}^\top \boldsymbol{\varphi} - \min_{\mathbf{q} \in \mathcal{U}(\mathbf{p}_t, d, \epsilon_t)} \mathbf{q}^\top \boldsymbol{\varphi} \right)$ is a continuous function on $\mathcal{A}$. It remains to be shown that the limit as $h \to 0$ of this function is also continuous.

First rewrite Equation 11 as the limit of functions indexed by positive integers $m \in \mathbb{Z}^+$ by defining $h = \frac{1}{m}$:

$$\hat{\delta}_t(\mathbf{x}) = \max_{\boldsymbol{\ell}_t(\mathbf{x}) \preceq \boldsymbol{\varphi} \preceq \mathbf{u}_t(\mathbf{x})} \zeta(\boldsymbol{\varphi})$$

$$\zeta := \lim_{m \to \infty} \zeta_m$$

$$\zeta_m(\boldsymbol{\varphi}) := m \left( \min_{\mathbf{q} \in \mathcal{U}(\mathbf{p}_t, d, \epsilon_t + \frac{1}{m})} \mathbf{q}^\top \boldsymbol{\varphi} - \min_{\mathbf{q} \in \mathcal{U}(\mathbf{p}_t, d, \epsilon_t)} \mathbf{q}^\top \boldsymbol{\varphi} \right)$$

By the uniform limit theorem, if each $\zeta_m$ is continuous and converges uniformly to $\zeta$, then $\zeta$ is continuous. Recall the definition of uniform convergence: $\zeta_m$ converges uniformly to $\zeta$ if for all $\eta > 0$, there exists an $M$ such that, for all $m \geq M$ and all $\boldsymbol{\varphi} \in \mathcal{A}$, $|\zeta(\boldsymbol{\varphi}) - \zeta_m(\boldsymbol{\varphi})| < \eta$.

We have previously shown that each $\zeta_m$ is continuous for any choice of convex distribution distance $d$. We now prove the uniform convergence of $\zeta_m$ when $d$ is the total variation (TV) distance.

For some $\boldsymbol{\varphi} \in \mathcal{A}$, consider a permutation $\boldsymbol{\sigma}$ of the entries of $\boldsymbol{\varphi}$ such that (breaking ties arbitrarily, without loss of generality)

$$\varphi_{\sigma_1} \geq \varphi_{\sigma_2} \geq ... \geq \varphi_{\sigma_n} .$$

To ease notation, for the rest of this proof, we drop the iteration $t$ subscript from the reference distribution $\mathbf{p}_t$ and the margin $\epsilon_t$. Apply the permutation $\boldsymbol{\sigma}$ to $\boldsymbol{\varphi}$ and the reference distribution $\mathbf{p}$ to get

$$\boldsymbol{\varphi}^\sigma := (\varphi_{\sigma_1}, \varphi_{\sigma_2}, ..., \varphi_{\sigma_n})$$
$$\mathbf{p}^\sigma := (p_{\sigma_1}, p_{\sigma_2}, ..., p_{\sigma_n})$$

Now define

$$\mathbf{q}_\epsilon^\sigma := \min_{\mathbf{q} \in \mathcal{U}(\mathbf{p}^\sigma, d, \epsilon_t)} \mathbf{q}^\top \boldsymbol{\varphi}^\sigma$$

Under this permutation of entries, $\mathbf{q}_\epsilon^\sigma$ will have the form prescribed in Lemma A.11. Note that we are solving an equivalent problem since the objective and constraint functions are all finite sums of vector entries (a feature of TV) and permuting them does not change their sums. As in Lemma A.11, let $k$ be the first index such that $\sum_{i=1}^k p_i^\sigma > \frac{\epsilon}{2}$. Then, noting that $k$ does not exist if $\sum_{i=1}^n p_i^\sigma \leq \frac{\epsilon}{2}$, define

$$c^\varphi := \begin{cases} \sum_{i=1}^k p_i^\sigma - \frac{\epsilon}{2}, & \text{if } k \text{ exists}, \\ \infty, & \text{otherwise}. \end{cases}$$

If $k$ exists, for any $m > \frac{1}{c^\varphi}$,

$$\zeta_m(\boldsymbol{\varphi}) = m(\mathbf{q}_{\epsilon + \frac{1}{m}}^\sigma - \mathbf{q}_\epsilon^\sigma)^\top \boldsymbol{\varphi}$$

$$= \frac{1}{h}(\mathbf{q}_{\epsilon + h}^\sigma - \mathbf{q}_\epsilon^\sigma)^\top \boldsymbol{\varphi}$$

$$\overset{(i)}{=} \frac{1}{h} \left( \left( \left( q_{\epsilon,k}^\sigma - \frac{h}{2} \right) \varphi_k^\sigma + \left( q_{\epsilon,n}^\sigma + \frac{h}{2} \right) \varphi_n^\sigma \right) - \left( q_{\epsilon,k}^\sigma \varphi_k^\sigma + q_{\epsilon,n}^\sigma \varphi_n^\sigma \right) \right)$$

$$= \frac{1}{2}(\varphi_n^\sigma - \varphi_k^\sigma)$$

where $(i)$ uses the closed form prescribed in Lemma A.11 and the fact that, for $h < c^\varphi$, the index $k$ stays the same. Since the final term is a constant with respect to $m$, we conclude that

$\zeta_m(\boldsymbol{\varphi}) = \lim_{i \to \infty} \zeta_i(\boldsymbol{\varphi}) = \zeta(\boldsymbol{\varphi})$ when $m > \frac{1}{c^{\boldsymbol{\varphi}}}$. If $k$ does not exist, for any $m > 0$, $\zeta_m(\boldsymbol{\varphi}) = \lim_{i \to \infty} \zeta_i(\boldsymbol{\varphi}) = \zeta(\boldsymbol{\varphi}) = 0$ since all terms in this sequence share the same form according to Lemma A.11.

Consider the index set $\{\sigma_1, \sigma_2, ..., \sigma_k\} \subseteq [n]$ (order does not matter). Every $\boldsymbol{\varphi} \in \mathcal{A}$ is associated with one such index set by the above construction. $\mathcal{A}$ can therefore be partitioned into a finite number of equivalence classes (specifically, $2^n$ classes), where each such equivalence class contains all $\boldsymbol{\varphi} \in \mathcal{A}$ with the same index set.

Observe that, for $\boldsymbol{\varphi}_1$ and $\boldsymbol{\varphi}_2$ in the same class, $c^{\boldsymbol{\varphi}_1} = c^{\boldsymbol{\varphi}_2}$, since $c^{\boldsymbol{\varphi}}$ only depends on $\mathbf{p}$, $\epsilon$, and the index set. Give each index set a meta-index $a \in [2^n]$, and let $c_a$ denote the value of $c^{\boldsymbol{\varphi}}$ shared by all $\boldsymbol{\varphi}$ in index set $a$. Now choose

$$M > \max_{a \in [2^n]} \frac{1}{c_a}$$

where the maximum is guaranteed to exist since all $c_a > 0$ and the maximization is over a finite set. For all $m \geq M$ and all $\boldsymbol{\varphi} \in \mathcal{A}$, $\zeta_m(\boldsymbol{\varphi}) = \zeta(\boldsymbol{\varphi})$ and thus $|\zeta_m(\boldsymbol{\varphi}) - \zeta(\boldsymbol{\varphi})| = 0$. We thus have uniform convergence, which completes the proof. □

## B EMPIRICAL STUDY OF TS-BOCU WITH WRONG HYPERPARAMETERS

We also study the performance of TS-BOCU on various uncertainty objectives when the hyperparameters are purposely misconfigured, i.e., when TS-BOCU is given values of $\alpha, \beta$ and $\epsilon$ that differ from that of the true uncertainty objective. The results are shown in Fig. 3 for the DRO, WCS, and GEN objectives (identical to those in Sec. 5), and TS-DRO (TS-BOCU with $\alpha = 1, \beta = 0$), TS-WCS ($\alpha = 0, \beta = 1$), and TS-GEN ($\alpha = 1, \beta = 1$), evaluated on all uncertainty objectives including those with different values of $\alpha, \beta$ and $\epsilon$. We first observe that TS-BOCU with the same hyperparameters as the true uncertainty objective generally performs close to the best, as expected. The next interesting observation is that, under the DRO and WCS objectives, TS-GEN always performs better than the other 'wrong' algorithm (i.e., better than TS-WCS under the DRO objective, and better than TS-DRO under the WCS objective). This empirically supports the interpretation that choosing values of $\alpha$ and $\beta$ to be both greater than 0 results in an algorithm that 'interpolates' between DRO ($\alpha = 1$) and RS ($\beta = 1$) objectives in the sense that it performs decently well on both and can be seen as 'robust' to the choice of uncertainty objective used to evaluate performance.

## C EXPERIMENTAL DETAILS

### C.1 EXPERIMENTAL SETTINGS

Supposing the decisions have dimension $m$ and the contexts have dimension $\ell$, all experiments have domain $[0, 1]^m \times [0, 1]^\ell$ by normalizing to this set from their original domains. We set the number of decisions $|\mathcal{X}| = 1024$, and the number of contexts $|\mathcal{C}| = n = 64$. The reference distribution $\mathbf{p}_t$ at all iterations is a Gaussian with mean $0.5 \cdot \mathbf{1}_\ell$ and covariance $0.2 \cdot \mathbf{I}_\ell$ (where $\mathbf{1}_\ell$ and $\mathbf{I}_\ell$ are a vector of ones and the identity matrix respectively), discretized into a probability vector of size $n$. The true distribution $\mathbf{p}_t^*$ at all iterations is a uniform distribution over $[0, 1]^\ell$ discretized into a probability vector of size $n$. The margin at all iterations is $d(\mathbf{p}_t, \mathbf{p}_t^*)$.

During the learning procedure, we use a GP with mean 0 and a ARD squared exponential kernel with $k(\mathbf{x}, \mathbf{c}; \mathbf{x}, \mathbf{c}) = 1$ and lengthscale 0.1 for each dimension. We set the observational noise standard deviation $\sigma = 0.01$, and the number of initial observations at the start of each learning procedure to be 5.

For TS-BOCU, we approximate sampling from the posterior using random Fourier features (Rahimi & Recht, 2007) with 1024 features. For all UCB algorithms, we set $\beta_t = 2$ for all iterations.

Samples from GP prior: $m = 2, \ell = 2$. The GP prior is the same as that used in the learning procedure described above.

Hartmann 3-D: $m = 2, \ell = 1$. The first two variables in the input are the decisions, while the remaining is the context. This function is described at `https://www.sfu.ca/~ssurjano/hart3.html`.

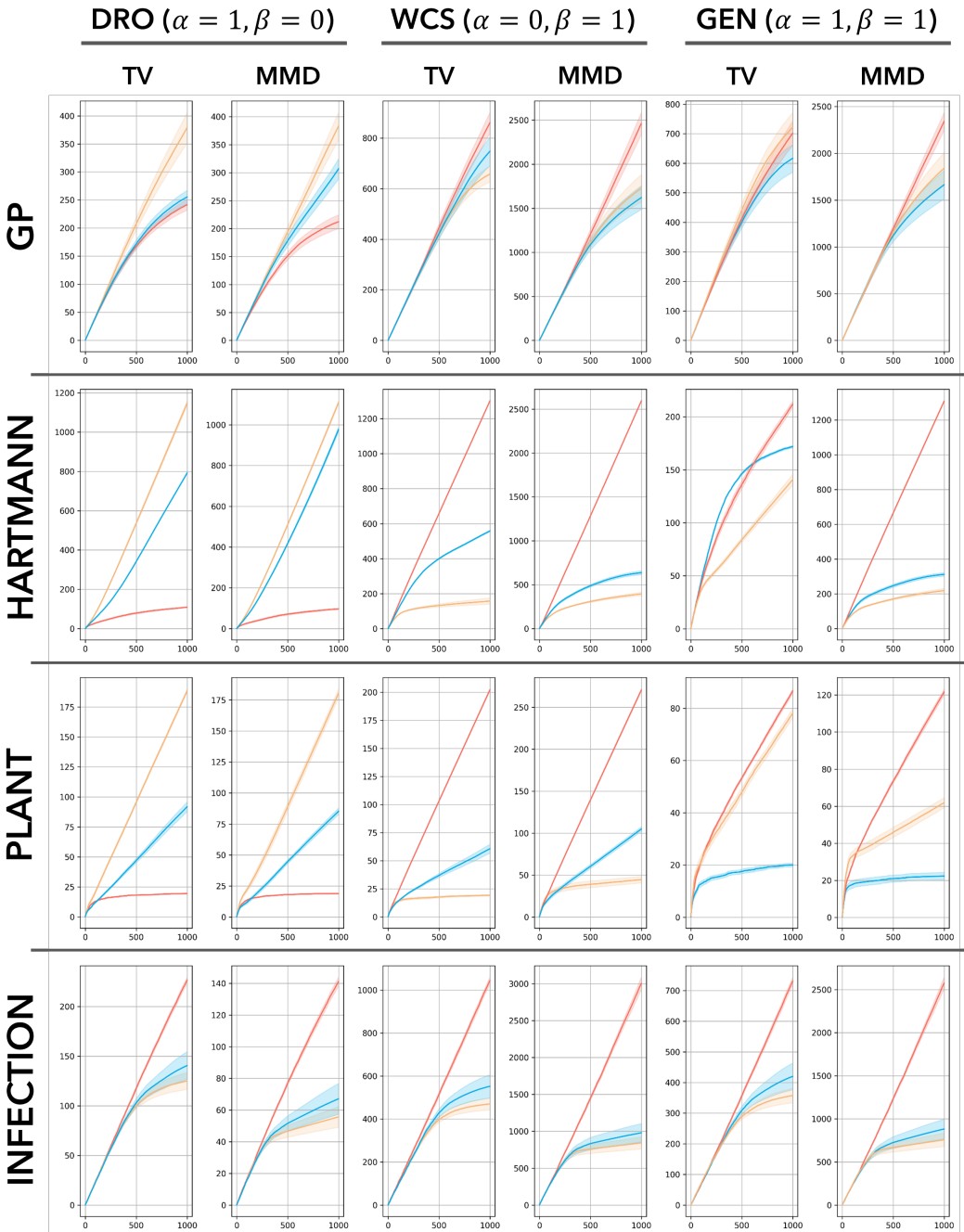

Figure 3: Mean and one standard error (over 10 RNG seeds) of cumulative regret (lower is better) incurred against iterations by the acquisitions **TS-DRO** (TS-BOCU with $\alpha = 1, \beta = 0$), **TS-WCS** ($\alpha = 0, \beta = 1$), **TS-GEN** ($\alpha = 1, \beta = 1$) with varying objective functions, uncertainty objectives, and distribution distances.

Plant growth simulator: $m = 1, \ell = 1$. This underlying function is the mean function of a heteroscedastic GP fit to private training data on the growth of *Marchantia* plants within a nutrient medium as a function of pH (with domain [2.5, 6.5]) and amount of ammonium $NH_3$ (with domain [0, 30] in mM). This underlying function is the same as that in Tay et al. (2022).

COVID-19 infection simulator: $m = 2, \ell = 3$. This underlying function is a function fit to COVID-19 simulator outputs with kernel ridge regression, using the same kernel as described above. The simulator was obtained from `https://github.com/peter-i-frazier/group-testing`. The decision variables are proportions of test kits allocated to 3 different sub-groups, the context variables are the initial number of cases within these sub-groups and the transmission probability, and the output is the number of resultant cases. This underlying function is the same as that in Tay et al. (2022). For the full list of simulator hyperparameters, refer to their code.

The experiments were implemented in Python using NumPy (Harris et al., 2020), PyTorch (Paszke et al., 2019), GPyTorch (Gardner et al., 2018) and BoTorch (Balandat et al., 2020). For full details, refer to the code repository at `<disclosed-on-publication>`.

## C.2 BASELINES

In this section, we provide details on the baselines UCB-BOCU-1 and UCB-BOCU-2. These baselines are naive extensions of the UCB algorithm for DRBO from Kirschner et al. (2020) that are computationally tractable but (as far as we can tell) have no theoretical guarantees on their performance.

UCB-BOCU-1:

$$U_t^{(1)}(\mathbf{x}) \coloneqq \alpha \hat{v}_t(\mathbf{x}) + \beta \hat{\delta}_t(\mathbf{x})$$

$$\hat{v}_t^{(1)}(\mathbf{x}) \coloneqq \min_{\mathbf{q} \in \mathcal{U}(\mathbf{p}_t, d, \epsilon_t)} \mathbf{q}^\top \mathbf{u}_t(\mathbf{x})$$

$$\hat{\delta}_t^{(1)}(\mathbf{x}) \coloneqq \frac{1}{\tilde{h}} \left( \min_{\mathbf{q} \in \mathcal{U}(\mathbf{p}_t, d, \epsilon_t + \tilde{h})} \mathbf{q}^\top \mathbf{u}_t(\mathbf{x}) - \min_{\mathbf{q} \in \mathcal{U}(\mathbf{p}_t, d, \epsilon_t)} \mathbf{q}^\top \boldsymbol{\ell}_t(\mathbf{x}) \right) .$$

At each iteration $t$, UCB-BOCU-1 chooses $\mathbf{x}_t = \arg\max_{\mathbf{x} \in \mathcal{X}} U_t^{(1)}(\mathbf{x})$. As compared to the true upper confidence bound $U_t$ described in the main paper, UCB-BOCU-1 replaces the maximization over all functions $\varphi$ in the derivative upper bound term with a difference between the worst-case expected values using the upper and lower confidence bounds of the function values. Note that the limit as $h \to 0$ is infinity if $\mathbf{u}_t(\mathbf{x}) \succ \boldsymbol{\ell}_t(\mathbf{x})$, and as such the limit is replaced with a small constant $\tilde{h} > 0$. Supposing that the derivative definition in the objective were similarly replaced with $\tilde{h}$, UCB-BOCU-1 actually uses a derivative upper bound that upper bounds the true derivative with high probability. However, the regret analysis that emerges from this choice leads to a factor of $\tilde{h}^{-1}$ on the regret upper bound. Since we tend to choose small values of $\tilde{h}$ for a good finite difference estimation of the derivative, this leads to a very large upper bound on the regret which explains the poor empirical performance.

UCB-BOCU-2:

$$U_t^{(2)}(\mathbf{x}) \coloneqq \alpha \hat{v}_t(\mathbf{x}) + \beta \hat{\delta}_t(\mathbf{x})$$

$$\hat{v}_t^{(2)}(\mathbf{x}) \coloneqq \min_{\mathbf{q} \in \mathcal{U}(\mathbf{p}_t, d, \epsilon_t)} \mathbf{q}^\top \mathbf{u}_t(\mathbf{x})$$

$$\hat{\delta}_t^{(2)}(\mathbf{x}) \coloneqq \lim_{h \to 0} \frac{1}{h} \left( \min_{\mathbf{q} \in \mathcal{U}(\mathbf{p}_t, d, \epsilon_t + h)} \mathbf{q}^\top \mathbf{u}_t(\mathbf{x}) - \min_{\mathbf{q} \in \mathcal{U}(\mathbf{p}_t, d, \epsilon_t)} \mathbf{q}^\top \mathbf{u}_t(\mathbf{x}) \right) .$$

At each iteration $t$, UCB-BOCU-2 chooses $\mathbf{x}_t = \arg\max_{\mathbf{x} \in \mathcal{X}} U_t^{(2)}(\mathbf{x})$. UCB-BOCU-2 replaces the maximization over all functions $\varphi$ in the derivative upper bound term by simply choosing the upper confidence bound function instead. Unlike UCB-BOCU-1, the limit exists and the value of the derivative upper bound converges. This possibly explains its better empirical performance. However, there is no theoretical guarantee that this choice of derivative upper bound has a high probability of being greater than or equal to the true derivative, and thus there is (as far as we can tell) no theoretical upper bound on the regret incurred by UCB-BOCU-2.

