# OpenReview forum: "A Unified Framework for Bayesian Optimization under Contextual Uncertainty"
_ICLR.cc/2024/Conference — ICLR 2024 poster_

### Official Review · Reviewer_WGc6 · 2023-10-28

**Soundness:** 3 good
**Presentation:** 3 good
**Contribution:** 3 good
**Rating:** 8
**Confidence:** 3

**Summary:**

Many problem settings in Bayesian optimization (BayesOpt) require robust solutions to the optimization problem, where the decision maker needs to choose solutions that work well under different contexts that are not controllable.
This paper presents a formulation that generalizes many different BayesOpt settings related to robust optimization, and proposes using Thompson sampling as the optimization policy.
The authors first show a regret bound of this policy that is sublinear given specific assumptions and subsequently demonstrate the empirical performance of the policy under various optimization settings.

Edit: After the rebuttal, I've increased my score from 6 to 8.

**Strengths:**

The paper is well-written and has a clear exposition.
I enjoyed reading through the formulation of the distributionally robust optimization problem and how it generalizes to previously proposed settings, especially by incorporating the right derivative of the objective.
This general framework allows relating many problem formulations that have been proposed in the literature.
As explained in the paper, by setting the hyperparameters, one can even realize novel optimization formulations that "interpolate" the previously proposed formulations at the extremes, which allows more expressiveness in designing optimization objectives that fit a user's preference.
The proposed TS seems to work well across the experiments.

**Weaknesses:**

The authors can consider inspecting why in various cases (in the Infection problem for DRO), TS-BOCU has almost linear regret.
It would be interesting to see if there are types of problems that the policy tends to perform badly on.
(Perhaps this is connected to the insight that the algorithm resulting from setting $\alpha, \beta > 0$ tends to be more robust?)

I am a bit confused about the assumption that $\mathcal{X}$ and $\mathcal{C}$ are finite: Is it necessary for the theoretical guarantee (since Section 4.1 mentions that the result can be extended to infinite sets)?
In the experiments (for example, with the Hartmann functions), are you constraining the search spaces to be finite?
My understanding is that  the algorithm can run on continuous search spaces too.

I think the paper can benefit from extending the discussion at the end of Section 3.1 and offer guidance for setting $\alpha$, $\beta$, and $\epsilon$ in practice.

**Questions:**

- Instead of having both $\alpha$ and $\beta$, can't we follow the formulation of mean-risk tradeoff and only vary the weight for the $\delta(\mathbf{x})$ term, unless, for example, $\alpha = \beta = 1$ gives a different objective than $\alpha = \beta = 0.5$ (assuming the same $\epsilon$)?
- As I understand, if $\epsilon_t = d(\mathbf{p}_t, \mathbf{p}^*)$ does not approach $0$ with probability $1$, we don't obtain the sublinear regret result.
How does this might affect performance in practice?
Can we still perform well if $\mathbf{p}_t$ is sufficiently different from $\mathbf{p}^*$?
Are there situations where $\mathbf{p}^*$ is very hard to learn?

---

> ### Author Response · Authors · 2023-11-15
>
> Thank you for the time and effort spent writing this review, and for your interesting questions.
>
> 1. **On the necessity of both** $\alpha$ **and** $\beta$: Both parameters are necessary, as robust satisficing and worst-case sensitivity have $\alpha=0$ and do not involve the $v(\mathbf x)$ term at all. If only $\beta$ were present, we would not be able to set the weight on the $v(\mathbf x)$ term to $0$ to recover these objectives.
>
> 2. **On what happens if** $d(\mathbf p_t, \mathbf p^*)$ **does not approach** $0$: In practice, even if $\mathbf p_t$ does not approach $\mathbf p^*$, we are still able to get good performance. In our experiments, for simplicity, we keep $\mathbf p_t$, $\mathbf p^*$, and $\epsilon_t > 0$ constants throughout, and we see that TS-BOCU still performs well and displays the sublinear property. The necessary condition for good performance is nuanced: informally speaking, we can perform well as long as the learner is able to reduce the uncertainty at all 'relevant' decision-context points, where 'relevance' is determined by the supports of the distributions within the uncertainty set $\mathcal U_t$. Notice that $\mathbf p^*$ only governs the distribution of contexts observed, and is not a part of the uncertainty objective: only the learner-chosen parameters $\mathbf p_t$ and $\epsilon_t$ are. This means that, if the true underlying function $f$ were completely known to the learner, it does not matter what $\mathbf p^*$ is, the learner would be able to solve the problem exactly. $\epsilon_t \rightarrow 0$ (with the constraint $d(\mathbf p_t, \mathbf p^*) \leq \epsilon_t$) is a sufficient but not necessary condition for good performance. $\mathbf p^*$ gets more difficult to learn as $|\mathcal C|$ increases.
>
> 3. **On the performance of TS-BOCU on the DRO uncertainty objective in Infection**: While the performance seems not great, we still observe the sublinear property towards the final iterations as the growth of the curve slows down. If we were to run the experiments for more iterations, this sublinear property would be more clear. Also, it may simply be the case that the baselines are very strong in this setting. UCB-BOCU is explicitly designed for the DRO uncertainty objective, whereas a possible reason that UCB-RO performs well is that the RO optimal decision coincides with the DRO optimal decision with this objective function. In any case, the sublinear property still assures us that our algorithm is able to handle the DRO objective as expected, although the relative performance between baselines will certainly vary depending on specific objective function.
>
> 4. **On the finite decision and context spaces**: We have formalized the extension to continuous, infinite decision sets in the new uploaded revision. The regret bound maintains the same order with respect to $T$, and the conditions necessary for sublinear regret remain the same as explained in the body of the paper. Please see our global response titled "Revision for infinite decision sets and intractability of infinite context sets" (https://openreview.net/forum?id=oMNkj4ER7V&noteId=EjvMtBW1sK) for more details. In that post, we have also defended the assumption of a finite context set. To summarize the salient points, DRO with infinite context sets is intractable with general convex distribution distances $d$, and this difficulty leads to the assumption of a finite context set being standard in the DRBO literature. Our experiment results are still from an implementation with a finite decision set, but we imagine that the results would not be too different if the optimization were performed over an infinite decision set instead.
>
> 5. **On guidance for setting** $\alpha$, $\beta$**, and** $\epsilon$ **in practice**: We agree with you that such a discussion would be interesting and useful. However, we also fear that such a discussion would become more subjective and philosophical than technical, and may depend too much on the individual use case. For example, what arguments could be made for a risk-averse learner to optimize the value-at-risk versus the conditional value-at-risk versus the mean-variance tradeoff? We decided in the end that an extended discussion on this topic would be out of the scope of this paper. We leave the choice of uncertainty objective to the practitioner; our goal was to show that the practitioner can choose between a large diversity of uncertainty objectives with a single mental framework and a single algorithm.
>
> We hope this response has improved your opinion of our paper.

---

> > ### Comment · Reviewer_WGc6 · 2023-11-17
> > **thanks**
> >
> > Thank you for the detailed response. The point about $\epsilon_t$ gives me new insight into the setting. It does seem reasonable that there are finitely many context variables $\mathbf{c}^{(i)}$.

---

### Official Review · Reviewer_B6kd · 2023-10-29

**Soundness:** 3 good
**Presentation:** 3 good
**Contribution:** 3 good
**Rating:** 5
**Confidence:** 4

**Summary:**

The paper studies robust BO when there is context uncertainty, and designs a TS algorithm based on a general framework that incorporate a large number of risk-sensitive learning objectives. The authors substantiate its efficacy with theoretical analysis and validate the algorithm's performance and adaptability through experiments.

**Strengths:**

The paper is clear and easy to follow. The propose method is solid and with decent theoretical and numerical evidence.

**Weaknesses:**

The significance is of question to me - though it is good to have a unified form for multiple previously proposed objectives, the unification achieved in this paper seems straightforward (adding two parameters) and the additional technical challenge (e.g., in algrotihm design or analysis) is unclear, and it is not clear whether those new objectives are really of significance for practitioners.

**Questions:**

1. Is the first-derivative objective only previously proposed for GP, or is also applicable in other areas? Is 3.1 only novel in GP literature, or for the first time also in other areas?

2. Why finite context and action space? These read very limited. Are they also required by prior works?

---

> ### Author Response · Authors · 2023-11-14
>
> Thank you for the time and effort taken to review our paper.
>
> 1. **On the significance of the unification and the analysis**: We believe that the significance of the unification goes beyond adding two parameters, as the right-derivative of the DRO objective at any $\epsilon$ was not known in the BOCU literature to be an uncertainty objective of interest. Tay et. al. (2022) introduced the idea of worst-case sensitivity to DRBO, but the worst-case sensitivity is only the right-derivative at $\epsilon = 0$. Our work is the first to establish the general usefulness of the right-derivative by proving in Proposition 3.1 that the right-derivative at $\epsilon > 0$ is of theoretical interest as well through the robust satisficing objective. We believe that this is a non-trivial and elegant extension of DRBO.  As you have pointed out, it is good to have a unified form for previously unrelated objectives, but the unification was anything but straightforward.
>
>     Recognizing and formally proving the relevance of the right-derivative was only the first part: the novel analysis of the regret bound for Thompson sampling when the right-derivative is part of the objective was not a straightforward undertaking. As we mention in the paper, the derivative term presents significant challenges: the limit term causes difficulties without a closed form, and the inner term is a difference of worst-case expected values which, informally speaking, `erases' the dependency on the uncertainty set and precludes bounding it in terms of the margin $\epsilon$ as was done for the DRO objective. We encourage you to explore the supplementary material, in particular Lemma A.3 and its auxiliary lemmas Lemma A.4, A.5, A.11, A.12, and A.13, to see the work involved in bounding the DRO first-derivative regret. For rigor, we also prove with Proposition 4.2 that the upper bound sequence $U_t$ used in the proof of the regret bound is well-defined.
>
> 2. **On the first-derivative objective**: Sections 2 and 3 on the general framework are general enough to be applicable outside of BO as well; note that the sequential learning setting and GPs are only introduced in Section 4. Proposition 3.1 is an original proposition that, to the best of our knowledge, has not been published anywhere else. It is entirely plausible that a similar result exists in the operations research literature that we are unaware of, and that we have simply re-discovered it. However, given that robust satisficing was only introduced this year (Long et. al., 2023), it seems unlikely.
>
> 3. **On the finite decision and context spaces**: We have **extended our results to continuous, infinite decision sets in the new uploaded revision**. The regret bound maintains the same order with respect to $T$, and the conditions necessary for sublinear regret remain the same as explained in the body of the paper. Please see our global response titled "Revision for infinite decision sets and intractability of infinite context sets" (https://openreview.net/forum?id=oMNkj4ER7V&noteId=EjvMtBW1sK) for more details. In that post, we have also defended the assumption of a finite context set. To summarize the salient points, DRO with infinite context sets is intractable with general convex distribution distances $d$, and this difficulty leads to the assumption of a finite context set being standard in the DRBO literature.
>
> We hope this response has improved your opinion of our paper. Please let us know if you have any more concerns, we are happy to engage further.

---

> ### Author Response · Authors · 2023-11-21
> **Reminder**
>
> Hello, a gentle reminder that the author-reviewer discussion period ends in about 2 days. If you have any remaining questions or concerns about our work after reading our response, please let us know so we may address them. Thanks!

---

### Official Review · Reviewer_9tD6 · 2023-10-31

**Soundness:** 3 good
**Presentation:** 1 poor
**Contribution:** 2 fair
**Rating:** 5
**Confidence:** 3

**Summary:**

A framework for Bayesian optimization under contextual uncertainty unifies various formulation of Bayesian optimization, including distributionally robust optimization, stochastic optimization, robust optimization, robust satisficing, worst-case sensitivity, and mean-risk tradeoff.  The authors provide theoretical analyses on regret bounds and experimental results to compare several Bayesian optimization algorithms.

**Strengths:**

* This work serves the comprehensive understanding of Bayesian optimization under contextual uncertainty.
* Paper is well-organized.

**Weaknesses:**

* Paper is hard to follow.  For example,

The sentence "While standard BO assumes that the learner has full control over all input variables to the objective function, in many practical scenarios, the learner only has control over a subset of variables (decision variables), while the other variables (context variables) may be randomly determined by the environment" is too complex.  There are two whiles in one sentence.

The sentence "We assume that, at every iteration, some reference distribution $\boldsymbol p$ is known that captures the learner's prior knowledge of the distribution governing $\boldsymbol c$" is grammatically wrong.

"a probability vector in $\mathbb{R}^n$" should be "a probability vector in $[0, 1]^n$" for readability and understandability.

I think there are other grammar and presentation issues.  Please revise your submission carefully.

* I think that some assumptions are too strong.  For example, the assumptions on finite sets of $\mathcal{X}$ and $\mathcal{C}$ are not practical.  Moreover, I do not understand why $\boldsymbol p$ is known at the beginning of the optimization.  This assumption is not practically meaningful.

* Since theoretical results are built on the assumptions on finite sets of $\mathcal{X}$ and $\mathcal{C}$, they are limited.

* Reasoning and justification behind experimental results are not appropriately provided.

**Questions:**

* Table 1 can have a column for the corresponding references.  It would help understand and compare diverse algorithms

* Could you explain the intuition and meaning of knowing $\boldsymbol p$ at the beginning?

* I am not sure that the ICLR paper format allows it, but the table captions should be located on top of the tables.

---

> ### Author Response · Authors · 2023-11-15
>
> Thank you for your review.
>
> 1. **On the finite decision and context spaces**: We have **extended our results to continuous, infinite decision sets** in the new uploaded revision. The regret bound maintains the same order with respect to $T$, and the conditions necessary for sublinear regret remain the same as explained in the body of the paper. Please see our global response titled "Revision for infinite decision sets and intractability of infinite context sets" (https://openreview.net/forum?id=oMNkj4ER7V&noteId=EjvMtBW1sK) for more details. In that post, we have also defended the assumption of a finite context set. To summarize the salient points, DRO with infinite context sets is intractable with general convex distribution distances $d$, and this difficulty leads to the **assumption of a finite context set being standard in the DRBO literature**. This should address your concerns about the strengths of these assumptions and the corresponding usefulness of our theoretical results.
> 2. **On the reference distribution $\mathbf p$**: We wish to emphasize that the reference distribution $\mathbf p$ **is a problem parameter chosen by the learner and captures the learner's prior belief of the context distribution, and is not the true distribution** $\mathbf p^*$. It is very possible in real life to obtain a reasonable prior distribution for random contexts, such as from expert knowledge or estimation from historical data. For instance, in the introduction's example of sunlight in farming, the amount of solar radiation is a quantity that has been tracked by meteorological organizations for decades (e.g., https://nsrdb.nrel.gov/). A reasonable reference distribution of solar radiation can be estimated from this data. If the learner has no expert knowledge or historical data, they can simply set the reference distribution at iteration 1 $\mathbf p_1$ to be the uniform distribution with a large margin $\epsilon_1$ to capture the uncertainty, then update their reference distributions $\mathbf p_t$ for $t>1$ based on the observed contexts over the learning procedure. This is the 'data-driven' setting in Kirschner et. al. (2020) described under 'Conditions for sublinear regret' in Sec. 4.
>
>     From a more technical perspective, it is not possible to even define the DRO problem without a reference distribution. Every problem definition under the BOCU framework needs some reference distribution to say anything useful, including BO for expected values (Toscano-Palmarin \& Frazier, 2022), risk-averse BO (Cakmak et al., 2020; Nguyen et al., 2021a;b), and prior work in distributionally robust BO (Kirschner et al., 2020; Nguyen et al., 2020; Tay et al., 2022).
> 3. **On the "reasoning and justification behind experimental results"**. We have to kindly ask you to elaborate on your concerns. What exactly about the design of the experiments do you not understand or agree with? Is it the choice of 1) objective functions, 2) uncertainty objectives, 3) distribution distances, 4) problem parameters, or 5) baselines? We believe we have made a reasonable effort to compare our proposed algorithm to suitable baselines across a diversity of objective functions, uncertainty objectives, and distribution distances. If there is anything that does not make sense to you, please specify them so that we can discuss them and improve our paper.
> 4. Your style suggestions have been noted. In the latest revision, we have moved the caption of Table 1 to the top. Due to ICLR's citation style that requires the name and date, including the references in Table 1 would cause it to be too cluttered. We have instead included equation numbers beside each uncertainty objective; the relevant references are always right before each equation.
>
> We hope these revisions and clarifications have improved your opinion of our paper. We look forward to hearing more from you.

---

> ### Author Response · Authors · 2023-11-21
> **Reminder**
>
> Hello, a gentle reminder that the author-reviewer discussion period ends in about 2 days. If you have any remaining questions or concerns about our work after reading our response, please let us know so we may address them. Thanks!

---

> > ### Comment · Reviewer_9tD6 · 2023-11-21
> >
> > It is hard to understand which part is updated (OpenReview does not provide the previous version now).  Could you highlight the changes in red or any colors?

---

> > > ### Author Response · Authors · 2023-11-22
> > >
> > > The changes within the main paper are now highlighted in blue.

---

> > > > ### Comment · Reviewer_9tD6 · 2023-11-22
> > > >
> > > > Thank you for your update.
> > > >
> > > > Some of my concerns are resolved, so I am slightly increasing the score.

---

### Official Review · Reviewer_Lem3 · 2023-11-02

**Soundness:** 3 good
**Presentation:** 3 good
**Contribution:** 3 good
**Rating:** 6
**Confidence:** 3

**Summary:**

The paper extends the distributionally robust Bayesian optimization (DRBO) framework to a more general framework called “BO under contextual uncertainty” (BOCU). BOCU targets the problem of maximizing some uncertainty objective that takes the context distribution into account. Example problems include worst-case sensitivity, mean-risk trade-offs, DRBO, robust satisficing. The paper develops a general Thompson sampling algorithm that can optimize any objective within the framework. The paper also derives Bayesian regret bound for their developed framework. Finally, some experiments are conducted to illustrate the sublinear regret properties of the framework.

**Strengths:**

+ The paper is very well written and easy to understand. The problem settings, the previous works, the concepts are all well described.
+ The paper tackles an interesting problem which is to unify different problems (with the same theme of BO under contextual uncertainty) into one framework. The paper also proposes a general method that can solves this unified framework with different objectives. Theoretical analysis is also conducted to guarantee the performance of the proposed method.
+ The proposed method seems to be sound and reasonable to me.
+ The experiments (though a bit limited) are also conducted in order to understand the behaviours of the proposed method and to confirm the theoretical analysis.

**Weaknesses:**

To me, the main weaknesses of the paper are in the experimental evaluation. I list in the below some weak points that I found from the experimental evaluation:
+ The problems used in the evaluation (GP, Hartmann 3, plant growth simulator, COVID epidemic model) have quite low dimensions, ranging from 2 to 5.
+ I found the analysis regarding the experiments could be further elaborated. Currently, there is only one paragraph explaining a lot of results in one figure (Figure 2), I have to think a lot in order to understand what the reported results convey.
+ For the COVID infection problem, I found the results of DRO are not too good. I’m just wondering what are the issues of these cases?

**Questions:**

Apart from my comments and questions in the Weaknesses section, the authors could answer the additional following questions:
+ In Theorem 4.1, are the assumptions used common assumptions used in this particular research topic? What are the implications of these assumptions? Is it possible for these assumptions to be occurred in practice?
+ Also, from Theorem 4.1, is the maximum information gain \gamma_T bounded as in the standard BO algorithms? Which kernels will guarantee this maximum information gain to be bounded?

---

> ### Author Response · Authors · 2023-11-15
>
> Thank you for your review and your insightful comments.
>
> 1. **On the assumptions in Theorem 4.1.**: Note that we have uploaded a new revision of the paper to extend our results to infinite decision sets, and in so doing require an additional assumption.
>
>     a. **Assumption 1, that the GP kernel is either the squared exponential kernel or the Matérn kernel with $\nu > 2$**: This assumption is required for the extension to infinite decision sets, see our global post for more details. It is a standard assumption for the discretization argument (Kandasamy et al., 2018; Srinivas et al., 2010). This assumption is a modeling choice and thus can certainly be true in practice, in fact, these are the most popular kernels used in practice.
>
> 	b. **Assumption 2, that the distribution distance $d$ is the TV distance**: This assumption is not a common assumption in the relevant literature, and was made in order to aid regret analysis due to the challenging derivative term in our novel objective function. As mentioned in the paper, the assumption that $d$ is the TV distance aids analysis considerably as TV admits a closed form for the optimal solution of the convex optimization problem Equation 2 (see Lemma A.11 in Appendix A.11). This assumption is also a modeling choice and thus can also be true in practice if the practitioner wishes it to be.
>
> 	c. **Assumption 3a, that for all** $t\leq T$, $d(\mathbf p_t, \mathbf p_t^*) \leq \epsilon_t$ : This is a standard assumption in DRBO (Kirschner et. al., 2020; Tay et. al., 2022). This assumption is used to ensure that the true distribution $\mathbf p_t^*$ is within the uncertainty set $\mathcal U_t$, so that we can bound the distance between $\mathbf p_t^*$ and the worst-case distribution $\mathbf q_t$ via a triangle inequality with the reference distribution $\mathbf p_t$. Informally speaking, we need $\mathbf p_t^*$ to be within some known distance from $\mathbf q_t$, otherwise the regret (computed with $\mathbf q_t$) cannot be related to the observed contexts (drawn from $\mathbf p_t^*$). In practice, while $\mathbf p_t$ and the margin $\epsilon_t$ are chosen by the learner, $\mathbf p_t^*$ is unknown and thus it is not possible to guarantee that it holds, other than with a trivially large $\epsilon_t$. This issue can be alleviated by choosing $\mathbf p_t$ that is likely to be close to $\mathbf p_t^*$ via expert knowledge or estimation from historical data, or by taking the 'data-driven' approach from Kirschner et. al. (2020) and adapting $\mathbf p_t$ to the observed contexts during the learning procedure.
>
> 	d. **Assumption 3b, that for all $t\leq T$ and all** $c \in [|\mathcal C|]$, $p^*_{t, c} \geq p_{\text{min}} > 0$. This assumption was also made in Theorem 4.2 in Inatsu et. al. (2022). We require this assumption again in order to aid regret analysis due to the challenging derivative term in our novel objective function. Intuitively, if a context has $0$ probability of being observed, the uncertainty at decision-context pairs with that context is very difficult to reduce. In practice, since $\mathbf p_t^*$ is not controllable, this may not hold. However, this problem may be alleviated by a careful choice of context domain such that it is unlikely that any of the contexts have $0$ probability of occuring.
>
> 2. **On the maximum information gain $\gamma_T$**: $\gamma_T$ is not bounded in $T$, but as long as $\gamma_T < \mathcal O(T)$, the overall regret bound is sublinear in $T$ and the algorithm converges to the optimal solution. Examples of kernels that fulfill this condition are the linear, squared exponential, and Matérn ($\nu > 1$) kernels. From Theorem 5 in Srinivas et. al. (2010), for the linear kernel, $\gamma_T = \mathcal O(d \log T)$; for the squared exponential kernel, $\gamma_T = \mathcal O((\log T)^{d+1})$; for the Matérn kernel with $\nu > 1$, $\gamma_T = \mathcal O(T^{d(d+1)/(2v+d(d+1))}(\log T))$, where in this context $d = m + \ell$ is the dimensionality of the joint decision-context space.
>
> 3. **On the performance of TS-BOCU on the DRO uncertainty objective in Infection**: While the performance seems not great, we still observe the sublinear property towards the final iterations as the growth of the curve slows down. If we were to run the experiments for more iterations, this sublinear property would be more clear. Also, it may simply be the case that the baselines are very strong in this setting. UCB-BOCU is explicitly designed for the DRO uncertainty objective, whereas a possible reason that UCB-RO performs well is that the RO optimal decision coincides with the DRO optimal decision with this objective function. In any case, the sublinear property still assures us that our algorithm is able to handle the DRO objective as expected, although the relative performance between baselines will certainly vary depending on specific objective function.
>
> We hope our response has improved your opinion of our paper.

---

> ### Author Response · Authors · 2023-11-21
> **Reminder**
>
> Hello, a gentle reminder that the author-reviewer discussion period ends in about 2 days. If you have any remaining questions or concerns about our work after reading our response, please let us know so we may discuss them. Thanks!

---

> > ### Comment · Reviewer_Lem3 · 2023-11-22
> > **Reply**
> >
> > Hi author(s),
> >
> > Thank you for the response. Your response have cleared majority of my concerns (except the comments regarding the low dimensions of the problems used in the paper). I can understand the contribution of the paper better now.

---

### Author Response · Authors · 2023-11-14
**Revision for infinite decision sets and intractability of infinite context sets**

Thank you all reviewers for your reviews.

Due to popular demand, we are pleased to announce that **we have extended our results to continuous, infinite decision sets in the new uploaded revision**, via an iterative discretization argument. In our previous draft, we had only mentioned the possibility of such an extension; now, we have formalized the necessary arguments to guarantee that the theoretical results hold for continuous, infinite decision sets. The regret bound maintains the same order with respect to $T$, and the conditions necessary for sublinear regret remain the same as explained in the body of the paper. We only require 1 additional assumption: that the kernel is the squared exponential kernel, or a Matérn kernel with $\nu > 2$. Under this assumption, the derivatives of GP sample paths $f$ satisfy $\mathbb P (\sup_{(\mathbf x, \mathbf c) \in \mathcal X \times \mathcal C} |\frac{\partial f(\mathbf x, \mathbf c)}{\partial x_i}| > J) \leq ae^{-(J/b)^2}$ for all $i \in \{1, ..., m \}$ and kernel-dependent constants $a$ and $b$ (Ghosal & Roy, 2006). This high probability bound on the derivatives is a standard requirement that enables the analysis of continuous decision sets via the iterative discretization argument (Kandasamy et al., 2018; Srinivas et al., 2010). We additionally use the assumption to guarantee the Lipschitz continuity of the GP posterior variance, a necessary property for the discretization argument in Thompson sampling that was omitted in previous work (see Remark A.2 in Appendix A.3). In any case, since this assumption is widely made in order to guarantee the sublinearity of the maximum information gain $\gamma_T$, and since these kernels are commonly used in practice, this assumption is a mild one.

On the assumption of a finite context set, we believe that this assumption is a reasonable one as **DRO with infinite context sets is intractable with general convex distribution distances $d$**.
With a finite context set, the DRO problem (with any convex distribution distance) is exactly solved as a convex optimization problem and the worst-case distributions are represented as probability vectors. With an infinite context set, it is not possible to exactly solve the DRO problem in general, and it is not even clear how to represent the worst-case distributions.
Due to this difficulty, **this assumption is standard in DRBO** and is explicitly made in Kirschner et. al. (2020), Tay et. al. (2022), and Inatsu et. al. (2022). Nguyen et. al. (2020) "assume distributional uncertainty in which the context distribution $P_0$ is known only through a limited set of its i.i.d samples $S_n = (w_1, ..., w_n)$", which is equivalent to having some discretization of the context set. Husain et. al. (2023) do not use this assumption, but their method is restricted to $f$-divergences and relies on Monte Carlo sampling of the contexts to only approximate the DRO solution.
Outside of BO, in DRO, Gotoh et. al. (2020) also use this assumption to derive the closed forms of worst-case sensitivity for many convex distribution distances. The worst-case sensitivity is a crucial link between DRO and the other uncertainty objectives described in the paper. The extension to different notions of risk such as variance, range, conditional value-at-risk etc. depends on the ability to solve the DRO problem for general convex distribution distances. It is not clear that these results are possible without this assumption.

We hope our revision and clarification will improve your opinions of our paper.

---

### Meta-Review · Area_Chair_s7Gg · 2023-12-07

**Metareview:**

The authors consider the setting of Bayesian optimization under contextual uncertainty -- a spin on contextual Bayesian optimization wherein there may be some uncertainty regarding the context, captured by a distribution over its plausible values. The authors formalize this problem setting and identify distributionally robust Bayseian optimization (which has been studied previously) as a special case. The authors provide a Thomspon sampling algorithm for problems in this framework, which they analyze theoretically and evaluate empirically.

The reviewers expressed excitement regarding the problem setting and the treatment of it by the authors. They also commended the authors for their clarity in presentation and overall design to their approach. I believe this would be a valuable contribution to the ICLR program.

The reviewers did not some, mostly minor, perceived weaknesses in the manuscript as submitted, mostly regarding motivation for some choices made by the authors in their presentation. However, these issues appear to have been mostly addressed sufficiently during the author response period. I encourage the authors to reflect upon this discussion when revising their manuscript.

**Justification For Why Not Higher Score:**

Although the reviewers came around on this paper after the discussion period, their overall enthusiasm is still relatively muted.

**Justification For Why Not Lower Score:**

Following the author response and reviewer discussion periods, the reviewers came to a general consensus that the paper was sufficiently improved/clarified as a result of the discussion with the authors to warrant acceptance.

---

### Decision · Program_Chairs · 2024-01-16

Accept (poster)